# *What if Tomorrow is the World Cup Final?* Counterfactual Time Series Forecasting with Textual Conditions

**Shuqi Gu**[1]  **Yongxiang Zhao**[1]  **Baoyu Jing**[2]  **Kan Ren**[1]

## Abstract

Time series forecasting has become increasingly critical in real-world scenarios, where future sequences are influenced not only by historical patterns but also by forthcoming events. In this context, forecasting must dynamically adapt to complex and stochastic future conditions, which introduces fundamental challenges in both forecasting and evaluation. Traditional methods typically rely on historical data or factual future conditions, while overlooking counterfactual scenarios. Furthermore, many existing approaches are restricted to simple structured conditions, limiting their ability to generalize to the real-world complexities. To address these gaps, we introduce the task of counterfactual time series forecasting with textual conditions, enabling more flexible and condition-aware forecasting. We propose a comprehensive evaluation framework that encompasses both factual and counterfactual settings, even in the absence of ground truth time series. Additionally, we present a novel text-attribution mechanism that distinguishes mutable from immutable factors, thereby improving forecast accuracy under sophisticated and stochastic textual conditions. The project page is at https://seqml.github.io/TADiff/.

## 1. Introduction

Time series forecasting plays a critical role in many real-world domains, including energy (Lai et al., 2018), climate (Jing et al., 2024b; 2021), healthcare (He et al., 2023; Chen et al., 2024), and finance (Gao et al., 2024). Recent advances span from innovations in model architectures (Zeng

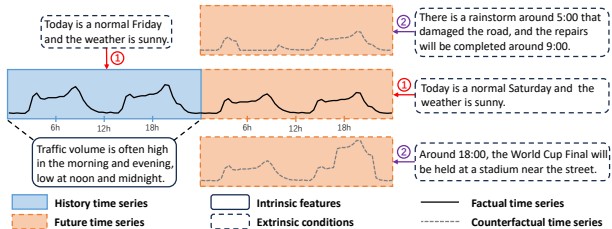

*Figure 1.* Two paradigms of time series forecasting in *traffic volume* with multimodal information are illustrated. Texts in the solid-line box describe the intrinsic features of the time series, whereas texts in the dashed-line box correspond to extrinsic conditions. ① denotes forecasting based on factual conditions, including the deterministic historical and future information. ② denotes forecasting under stochastic counterfactual conditions.

et al., 2023; Wu et al., 2021; Nie et al., 2022; Yuan & Qiao, 2024) to paradigm shifts enabled by large-scale learning with increasingly powerful models (Ansari et al., 2024; Woo et al., 2024). Nevertheless, despite larger model sizes and access to vast datasets, gains in forecasting performance have begun to plateau (Xu et al., 2024). A fundamental limitation of existing approaches is their exclusive reliance on historical observations, overlooking that information encoded in past trajectories is inherently limited and thus insufficient to generate reliable forecasts in hypothetical scenarios.

Time series are shaped by extrinsic conditions beyond historical sequences. Motivated by this, recent studies incorporate multimodal information, such as domain knowledge (Jin et al., 2023), contemporaneous news (Liu et al., 2024b), or expert annotations (Liu et al., 2024a; Lan et al., 2026) to enrich forecasting. While these extrinsic signals provide valuable supplementary context, they are still tied to historical states and thus cannot anticipate the influence of future conditions. When future dynamics deviate significantly from historical patterns, the forecasting of such models deteriorates. To mitigate this, Xu et al. (2024) proposes leveraging deterministic future interventions for improved forecasting. However, the future is inherently uncertain and can unfold differently under diverse conditions. Considering this uncertainty, Melnychuk et al. (2022) and Mu et al. (2025) model forecasts under counterfactual treatments, but these treatments are restricted to simple fixed event types, such as

[1]School of Information Science and Technology, ShanghaiTech University, Shanghai, China [2]Meta Platform Inc., California, United States. Correspondence to: Kan Ren <renkan@shanghaitech.edu.cn>.

*Proceedings of the 43rd International Conference on Machine Learning*, Seoul, South Korea. PMLR 306, 2026. Copyright 2026 by the author(s).

0/1 to indicate whether treatments are applied, which limits the flexibility to represent the real-world complexities that cannot be well formulated through categories.

To move beyond these constraints, we aim to answer a broader "*what if*" question: how might time series evolve dynamically under complex and stochastic future conditions, even when the conditions are not actually realized in the real world? We formalize this as *counterfactual conditional time series forecasting*, where future conditions are expressed through textual descriptions. Texts are more flexible to capture subtle details beyond categorical attributes (Gu et al., 2025) and align more naturally with human communication, thereby reducing the burden of manual categorical variable design. As shown in Fig. 1, the evolution of future traffic time series depends jointly on historical data and stochastic future events. This formulation introduces several key challenges. First, historical dynamics and future conditions may exhibit distinct or even conflicting patterns, which can lead to imbalanced forecasting that fails to account for their combined influence. Second, models trained solely on real-world data often struggle to adapt to counterfactual conditions, resulting in homogenized forecasts. Third, the lack of ground-truth for counterfactual settings presents a fundamental obstacle to evaluate forecasting quality.

To tackle these challenges, we propose the Text-Attributive time series Diffusion model (TADIFF), a multimodal diffusion model built upon a novel text-attribution mechanism for counterfactual time series forecasting. Specifically, since future sequences are jointly guided by future conditions and historical patterns, TADIFF first attributes historical sequences to intrinsic features that are independent of the historical and future conditions. These intrinsic features provide a foundation for integrating future conditions, enabling more accurate and condition-aware forecasting. To further improve adaptability, we introduce counterfactual data augmentation by synthesizing diverse counterfactual training samples. For evaluation, we design a novel semantic metric that measures the consistency of forecasts with both the textual conditions and the historical sequences, allowing comprehensive assessment even in counterfactual settings.

Our main contributions are: (i) We introduce a novel task, counterfactual time series forecasting with textual conditions, enabling accurate, flexible forecasting under stochastic assumptions. (ii) We propose a comprehensive evaluation framework that combines numerical accuracy and semantic alignment measures, allowing systematic benchmarking in both factual and counterfactual settings. (iii) We develop a text-attributive time series diffusion model (TADIFF) that disentangles intrinsic historical patterns from textual context, along with a training strategy that improves the model's adaptability to diverse counterfactual conditions through the constructed counterfactual data.

## 2. Related Work

### 2.1. Unimodal and Multimodal Time Series Forecasting

Time series forecasting typically involves predicting future values based on historical data. Research has focused on improving this by developing model architectures that better capture temporal patterns, including linear models (Zeng et al., 2023), CNNs (Wu et al., 2022), and transformer-based approaches (Wu et al., 2021; Nie et al., 2022). Recently, foundation models (Ansari et al., 2024; Woo et al., 2024; Feng et al., 2026) have emerged to learn universal time series representations from large datasets and model capacity. However, these methods still mainly rely on numerical historical sequences, overlooking the crucial impact of extrinsic conditions.

To address this limitation, recent studies have explored incorporating multimodal extrinsic information into forecasting. Among various modalities, text has become the most prevalent, appearing as domain knowledge (Jin et al., 2023), news (Liu et al., 2024b; Wang et al., 2024), or expert annotations (Liu et al., 2024a), providing complementary signals for prediction. Other approaches transform time series into alternative modalities, such as frequencies (Li et al., 2025) or images (Zhong et al., 2025), to enhance model expressiveness. Nonetheless, these methods predominantly derive multimodal inputs from historical data, limiting their capacity to address scenarios where future dynamics diverge substantially from past observations.

More recent works (Xu et al., 2024; Ashok et al., 2025) have begun to incorporate future interventions into forecasting, with Williams et al. (2025) introducing a benchmark for evaluating time series forecasting conditioned on future events. While these studies recognize the importance of modeling future conditions, they fail to consider the counterfactual setting, where interventions and events are described deterministically, failing to capture the inherent uncertainty of the future. Consequently, these forecasting methods tend to lack robustness and struggle to adapt to diverse counterfactual futures, as empirically demonstrated in Sec. 4.2.

### 2.2. Counterfactual Modeling

Since the real world is inherently uncertain, researchers are not satisfied with modeling only observed events; instead, they strive to uncover the underlying principles governing system behavior under counterfactual conditions. Such conditions manifest differently across domains. In reasoning tasks, prior work (Gendron et al., 2024) attempts counterfactual logical reasoning in natural language, while Wu et al. (2024) studies how events evolve under different treatments regarding the same time period. In generation and editing tasks, Rasal et al. (2025) generates non-existent images by altering specific features, whereas Jing et al. (2024a) modi-

fies attributes of source time series to produce a target one.

In this work, we focus on forecasting, which presents unique challenges compared to reasoning and generation due to the temporal dependencies involved. In forecasting, the joint influence of historical patterns and future extrinsic conditions should be considered organically. Most existing counterfactual forecasting approaches (Wu et al., 2024; Melnychuk et al., 2022; Mu et al., 2025) attempt to forecast multiple plausible futures under alternative treatments. However, they rely on structured attribute conditions, where possible futures are represented as fixed categories. In contrast, our work leverages unstructured textual conditions, enabling richer and more flexible conditional modeling with broader applicability to real-world scenarios. Furthermore, prior evaluation practices are constrained to synthetic datasets or limited to the observed conditions in real datasets, since counterfactual settings lack ground-truth futures. To address this gap, we propose a novel evaluation metric that assesses the consistency of forecasts with both historical sequences and counterfactual textual conditions, while remaining robust to the absence of ground truth.

## 3. TADIFF: Text-Attributive Time Series Diffusion Model

### 3.1. Counterfactual Time Series Forecasting

Given a sample of historical time series $\mathbf{x}_h \in \mathbb{R}^{L_h}$ with $L_h$ time steps, corresponding historical condition $\mathbf{c}_h \in \mathbb{N}^W$ in text with $W$ tokens, and $M$ potential future conditions $\{\mathbf{c}_f^{(1)}, \cdots, \mathbf{c}_f^{(M)}\}$ including the factual condition $\mathbf{c}_f^{(1)}$ and diverse counterfactual conditions $\{\mathbf{c}_f^{(2)}, \cdots, \mathbf{c}_f^{(M)}\}$, where each condition $\mathbf{c}_f^{(i)}$ is text describing the future $L_f$ time steps. The factual forecasting is conditioned on the combinations of historical sequence $\mathbf{x}_h$ and future condition $\mathbf{c}_f^{(i)}$ are sampled from the real world, while the counterfactual forecasting is conditioned on the unseen combinations in the real world. We aim to learn a forecasting model $G$, predicting the corresponding future time series $\hat{\mathbf{x}}_f^{(i)} = G(\mathbf{x}_h, \mathbf{c}_h, \mathbf{c}_f^{(i)}) \in \mathbb{R}^{L_f}$, where the forecasts should balance the impact of both history and future. Our target is to make the forecasting results $\hat{\mathbf{x}}_f^{(i)}$ follow the constraint of the historical sequence $\mathbf{x}_h$ and consistent with the future condition $\mathbf{c}_f^{(i)}$. With the ground truth $\mathbf{x}_f^{(i)}$, we further hope the forecasts $\hat{\mathbf{x}}_f^{(i)}$ is close to the ground truth $\mathbf{x}_f^{(i)}$.

### 3.2. Diffusion Model

Time series forecasting is inherently uncertain. This uncertainty arises from two sources: (i) the variability of future conditions, and (ii) the stochastic nature of time series evolution even under given conditions. To better capture this

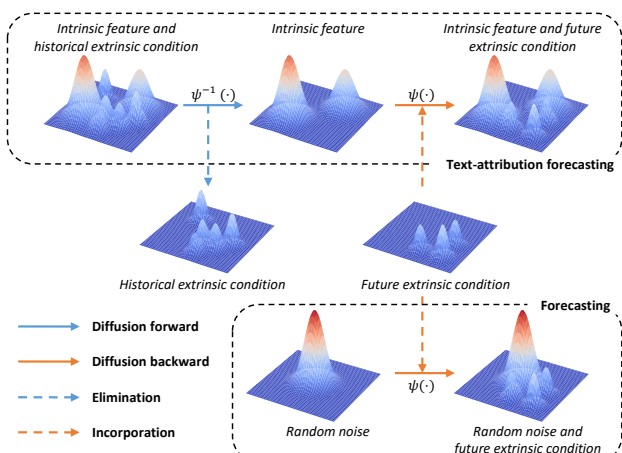

*Figure 2.* The visualization of the initial noise issue and our solution through text-attribution. The distribution plots represent the distribution of the time series under *specific influences*.

uncertainty, we adopt a generative modeling approach as the foundation of our forecasting framework. In particular, we employ diffusion models due to training stability and strong capacity for distributional modeling. Our forecasting model is built upon denoising diffusion implicit models (Song et al., 2020). Below, we provide a brief overview of the training and inference procedures of the diffusion model.

During training, noise is gradually added to the original data distribution $q(\mathbf{x}_0)$[1] via a Gaussian Markov transition:

$$q(\mathbf{x}_{1:T}|\mathbf{x}_0) = \prod_{t=1}^{T} q(\mathbf{x}_t|\mathbf{x}_{t-1}),$$
$$q(\mathbf{x}_t|\mathbf{x}_{t-1}) = \mathcal{N}(\sqrt{1-\beta_t}\mathbf{x}_{t-1}, \beta_t\mathbf{I}), \quad (1)$$

and produces the noisy sample $\mathbf{x}_t$ at each diffusion step $t \in [1, T]$. Here $\{\beta_t\}_{t=1}^{T}$ are the predetermined variance schedule. $\mathbf{x}_t$ can be expressed as $\mathbf{x}_t = \sqrt{\alpha_t}\mathbf{x}_0 + \sqrt{1-\alpha_t}\epsilon$, where $\epsilon \sim \mathcal{N}(\mathbf{0}, \mathbf{I})$ and $\alpha_t := \prod_{s=1}^{t}(1-\beta_s)$. Then, a learnable noise estimation network $\epsilon_\theta(\mathbf{x}_t, t, \mathbf{c})$ is trained by estimating the noise added to $\mathbf{x}_t$ given the condition $\mathbf{c}$ and diffusion step $t$. The objective function is to minimize the noise estimation loss as:

$$\min_\theta \mathcal{L}(\mathbf{x}_0) = \min_\theta \mathbb{E}_{\epsilon \sim \mathcal{N}(\mathbf{0},\mathbf{I}), t \sim \mathcal{U}(1,T)} \|\epsilon - \epsilon_\theta(\mathbf{x}_t, t, \mathbf{c})\|_2^2, \quad (2)$$

where $\mathbf{x}_0 \sim q(\mathbf{x}_0)$ is sampled from the real data distribution, $\mathbf{x}_t$ is a noisy version of $\mathbf{x}_0$.

During the inference phase, given a condition $\mathbf{c}$ and noisy data $\hat{\mathbf{x}}_t$, the denoising transitions $\psi_t(\cdot)$ to get less noisy data

---

[1]$\mathbf{x}$ and $\mathbf{x}_0$ are interchangeable in this paper.

$\hat{\mathbf{x}}_{t-1}$ can be formulated as:

$$\hat{\mathbf{x}}_{t-1} = \psi_t(\hat{\mathbf{x}}_t, \mathbf{c}) = \sqrt{\alpha_{t-1}}\hat{\mathbf{x}}_0 + \sqrt{1 - \alpha_{t-1}}\epsilon_\theta(\hat{\mathbf{x}}_t, \mathbf{c}, t),$$
$$\hat{\mathbf{x}}_0 = \frac{1}{\sqrt{\alpha_t}}(\hat{\mathbf{x}}_t - \sqrt{1 - \alpha_t}\epsilon_\theta(\hat{\mathbf{x}}_t, \mathbf{c}, t)).$$
$$(3)$$

This process can also be inverted to $\psi_t^{-1}(\cdot)$, which inversely estimates $\hat{\mathbf{x}}_{t+1}$ given $\hat{\mathbf{x}}_t$ and $\mathbf{c}$:

$$\hat{\mathbf{x}}_{t+1} = \psi_t^{-1}(\hat{\mathbf{x}}_t, \mathbf{c}) = \sqrt{\alpha_{t+1}}\hat{\mathbf{x}}_0 + \sqrt{1 - \alpha_{t+1}}\epsilon_\theta(\hat{\mathbf{x}}_t, \mathbf{c}, t).$$
$$(4)$$

**Discussion: Initial noise issue.** Most diffusion models (Yuan & Qiao, 2024; Gu et al., 2025) use Gaussian-distributed random noise $\hat{\mathbf{x}}_T \sim \mathcal{N}(0, \mathbf{I})$ as the initial state during inference. However, this randomly sampled noise is not optimal, as it may introduce properties that conflict with the intrinsic features of clear data $\mathbf{x}_0$, thereby complicating the denoising process, as shown in Fig. 2. To address this, we propose a condition-aware forward process to attribute the intrinsic features of history, leading to more accurate and controllable forecasting, which will be further discussed in Sec. 3.3. We further prove that our method optimizes the initial noise space of the diffusion model in Sec. 4.3.

## 3.3. Text-Attribution Mechanism

For counterfactual forecasting, there may be pattern conflicts between future conditions and historical sequences, such as the impact of the World Cup Final on traffic volume. This leads to difficulty for the forecasts to balance their combined effects. We argue that forecasts should first respect the intrinsic features of the historical sequence, which refer to the fundamental properties of the time series that remain unaffected by extrinsic conditions, such as the demographic structure underlying traffic series. This view is consistent with Leibniz's Principle of Continuity (Leibniz, 2012), which posits that natural changes occur gradually. At the same time, forecasts must faithfully capture the semantics embedded in future conditions, such as the future weather impact on the traffic series. To achieve balanced forecasting considering the combined effects of history and future, we propose a text-attribution mechanism that attributes historical sequences prior to forecasting, aiming to decouple the intrinsic features of the sequence from the extrinsic conditions. As shown in Fig. 3, the overall process consists of a two-stage inference and a joint training with two optimization objectives. We provide the details of the model architecture in Appendix C.

**Inference Stage 1: Attribution.** We believe that the historical sequence $\mathbf{x}_{h,0}$ is the result produced by the joint influence of the extrinsic condition $\mathbf{c}_h$ and the intrinsic feature. In our work, the attribution is achieved through a condition-aware diffusion forward process, where we diffuse the clear sequence $\mathbf{x}_{h,0}$ to its intrinsic features $\mathbf{x}_{h,T}$.

The influence of historical extrinsic condition $\mathbf{c}_h$ is gradually eliminated from historical sequence $\mathbf{x}_{h,0}$ through the condition-aware diffusion process:

$$\mathbf{x}_{h,T} = (\psi_{T-1}^{-1} \circ \cdots \circ \psi_0^{-1})(\mathbf{x}_{h,0}, \mathbf{c}_h), \qquad (5)$$

where $\circ$ represents the function composition, $\psi_t^{-1}$ is the inverse transition defined in Eq. 4. We provide the independence proof between $\mathbf{x}_{h,T}$ and $\mathbf{c}_h$ in Appendix E.3.

**Inference Stage 2: Forecasting.** As shown in the right half of Fig. 3, we take the intrinsic feature of historical sequence $\mathbf{x}_{h,T}$ as the initial state $\hat{\mathbf{x}}_{f,T}$ for forecasting:

$$\hat{\mathbf{x}}_{f,T} = \mathbf{x}_{h,T},$$
$$\hat{\mathbf{x}}_0 = \mathbf{x}_{h,0} \oplus \hat{\mathbf{x}}_{f,0} = (\psi_1 \circ \cdots \circ \psi_T)(\mathbf{x}_{h,0} \oplus \hat{\mathbf{x}}_{f,T}, \mathbf{c}_f),$$
$$(6)$$

where $\oplus$ is the concatenation operation along the time dimension, $\psi_t$ is the denoising process of diffusion model defined in Eq. 3. For each denoising step $t$, the first $L_h$ time points of the output from the previous step $t+1$ are replaced with the clear historical sequence $\mathbf{x}_{h,0}$. We take the last $L_f$ time steps of output $\hat{\mathbf{x}}_0$ as the forecasts.

**Forecasting Optimization.** We try to enhance the model forecasting given the historical sequence $\mathbf{x}_{h,0}$ and future condition $\mathbf{c}_f$. As illustrated in the left half of Fig. 3, the input time series for the diffusion noise estimator can be expressed as: $\mathbf{x}_t = \mathbf{x}_{h,0} \oplus \left[\sqrt{\alpha_t}\mathbf{x}_{f,0} + \sqrt{1 - \alpha_t}\epsilon\right] \in \mathbb{R}^{L_h + L_f}$, which is the mixed sequence containing the clear history and noisy future sequence, $t$ is the diffusion step. The loss function can be written as:

$$\mathcal{L}_F(\theta) = \mathbb{E}_{\epsilon \sim \mathcal{N}(\mathbf{0}, \mathbf{I}), t \sim \mathcal{U}(1, T)} ||\mathbf{m} \times [\epsilon_\theta(\mathbf{x}_t, \mathbf{c}_f, t) - \epsilon]||_2^2,$$
$$(7)$$

where $\mathbf{m} \in \{0\}^{L_h} \oplus \{1\}^{L_f}$ is the mask to make sure only the future components are supervised.

**Attribution Optimization.** The attribution is optimized through better noise estimation given the noisy historical sequence $\mathbf{x}_{h,t}$ and its corresponding condition $\mathbf{c}_h$. The loss is defined as the expectation of noise estimation error:

$$\mathcal{L}_A(\theta) = \mathbb{E}_{\epsilon \sim \mathcal{N}(\mathbf{0}, \mathbf{I}), t \sim \mathcal{U}(1, T)} ||\epsilon_\theta(\mathbf{x}_{h,t}, \mathbf{c}_h, t) - \epsilon||_2^2, \quad (8)$$

where $\mathbf{x}_{h,t} = \sqrt{\alpha_t}\mathbf{x}_{h,0} + \sqrt{1 - \alpha_t}\epsilon$ is the noisy historical time series.

**Overall Learning Objective.** The final loss function is the weighted sum of the two losses: $\mathcal{L}(\theta) = \lambda_F \cdot \mathcal{L}_F(\theta) + \lambda_A \cdot \mathcal{L}_A(\theta)$, where the forecasting and attribution share the same model parameters $\theta$.

**Advantages.** Compared with the generation method (Gu et al., 2025), we consider the influence of intrinsic features from historical sequences. Compared with forecasting diffusion models (Su et al., 2025; Yuan & Qiao, 2024), the

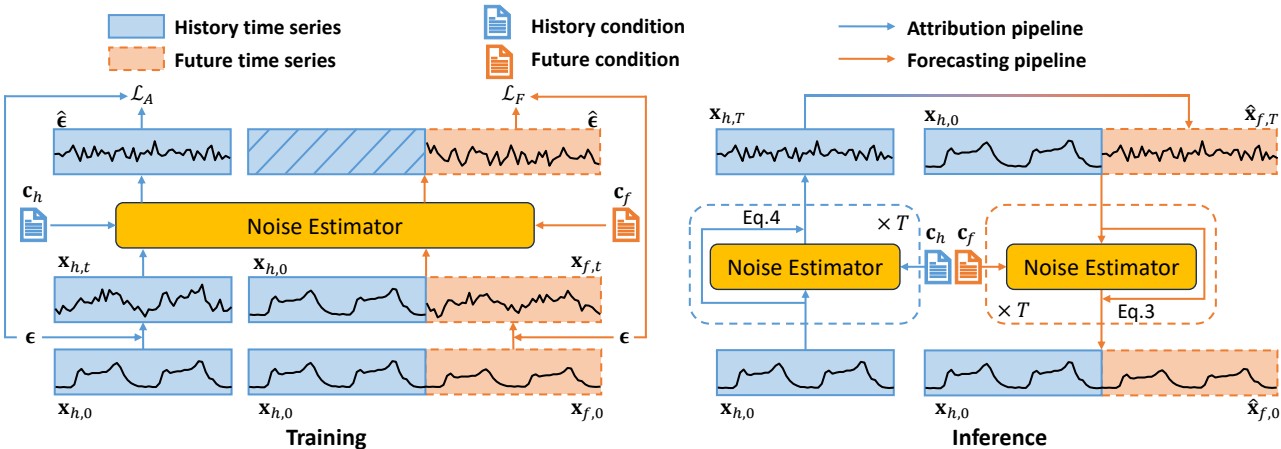

*Figure 3.* The framework of TADIFF, including the joint optimization of attribution and forecasting, as well as the two-stage inference. TADIFF first attributes the intrinsic features $\mathbf{x}_{h,T}$ of the historical sequence $\mathbf{x}_{h,0}$ given the historical textual condition $\mathbf{c}_h$. Then, conditioned on the historical sequence $\mathbf{x}_{h,0}$ and the future textual condition $\mathbf{c}_f$, TADIFF forecasts the future sequence $\hat{\mathbf{x}}_{f,0}$ by leveraging the intrinsic historical features $\hat{\mathbf{x}}_{f,T} = \mathbf{x}_{h,T}$ as initial noise state of diffusion.

attribution balances the combined influence between the history sequences and future conditions. As proved in Sec. 4.3, building conditional forecasts on top of these attribution results leads to more accurate and rational predictions.

### 3.4. Evaluate Forecasts without Ground Truth

Evaluating forecasts under counterfactual conditions is inherently challenging since the ground truth of future sequences is absent. Conventional accuracy-based metrics (i.e., mean square error) are only applicable in real-world scenarios, failing to assess counterfactual forecasting, underscoring the need for novel evaluation. We believe the forecasts are shaped by two key factors: intrinsic features of the historical sequences and extrinsic control of the future conditions. To evaluate the influences of the two factors, we introduce a semantic evaluation metric, DTTC (**D**isentangled **T**ime Series and **T**ext **C**onsistency), which consists of DTTC-I and DTTC-E, measuring the consistency of forecasts with intrinsic historical features and extrinsic future conditions, respectively. This approach builds on the observation that a time series embodies both intrinsic features and condition-dependent characteristics, while the textual conditions play the role of extrinsic control.

The DTTC model contains two encoders: a disentangled time series encoder $E_{\text{ts}}$ and a text encoder $E_{\text{text}}$. The time series encoder $E_{\text{ts}}$ disentangles the time series into the intrinsic and extrinsic features, which are applied to both the historical sequence $\mathbf{I}_h$, $\mathbf{E}_h = E_{\text{ts}}(\mathbf{x}_h)$ and future sequence $\mathbf{I}_f$, $\mathbf{E}_f = E_{\text{ts}}(\mathbf{x}_f)$. The text encoder $E_{\text{text}}$ embeds the textual conditions into the extrinsic features, which are applied to both the historical condition $\tilde{\mathbf{E}}_h = E_{\text{text}}(\mathbf{c}_h)$ and future condition $\tilde{\mathbf{E}}_f = E_{\text{text}}(\mathbf{c}_f)$. We aim to align the intrinsic features between historical and future sequences, as well

as the extrinsic features between sequences and their corresponding texts. These alignments are realized through contrastive learning (Radford et al., 2021). These feature embeddings are used to calculate the DTTC-I and DTTC-E:

$$\text{DTTC-I} = \frac{1}{N}\sum_{i=1}^{N} <\mathbf{I}_f[i], \mathbf{I}_h[i]>;$$
$$\text{DTTC-E} = \frac{1}{N}\sum_{i=1}^{N} <\mathbf{E}_f[i], \tilde{\mathbf{E}}_f[i]>, \tag{9}$$

where $< \cdot, \cdot >$ denotes the inner product, $N$ is the sample number. DTTC-I assesses the consistency between forecasts and the intrinsic historical features, while DTTC-E evaluates the consistency between forecasts and the future conditions.

The training of the DTTC model follows a dataset-specific setting, meaning that the model is independently trained on the training split of each dataset using contrastive learning. Additional training and model details are provided in Appendix B. We further evaluate the DTTC model on the test split of each dataset to prove that DTTC-I and DTTC-E are reliable metrics for counterfactual forecasting, as the evaluation results are reported in Tab. 4 of Sec. 4.3.

Compared with prior studies (Melnychuk et al., 2022; Mu et al., 2025), which rely solely on synthetic data for counterfactual evaluation, our approach enables the assessment of counterfactual forecasting on real-world datasets even without ground-truth time series. Rather than focusing on the accuracy of predicted values, our metrics prioritize the rationality and semantic consistency of the forecasts.

## 3.5. Adapt to diverse future conditions

Training solely on the factual data makes the models struggle to adapt to dynamic counterfactual conditions, since each historical sequence corresponds to a unique and deterministic future condition in real-world datasets. To address this challenge, we propose a finetuning algorithm based on constructed counterfactual future conditions, designed to enhance the model's adaptability in forecasting under diverse conditions, even in the absence of future ground truth.

**Dataset Construction.** Our factual data consists of a quadruple of historical and future time series and conditions: $\mathcal{D} = \{(\mathbf{x}_h, \mathbf{c}_h, \mathbf{x}_f, \mathbf{c}_f)_i\}_{i=1}^N$, where $N$ is the number of samples. We first construct a counterfactual dataset by randomly assigning alternative future conditions to each historical sequence. Specifically, for each historical time series $\mathbf{x}_h$ with its associated condition $\mathbf{c}_h$, we randomly synthesize $M$ candidate future conditions $\{\tilde{\mathbf{c}}_f^{(1)}, \cdots, \tilde{\mathbf{c}}_f^{(M)}\}$ from the factual dataset by sampling. Among them, we select the condition $\tilde{\mathbf{c}}_f^{(k)}$ that achieves the highest similarity score with $\mathbf{c}_h$: $k = \arg\max_k < E_{\text{text}}(\mathbf{c}_h), E_{\text{text}}(\mathbf{c}_f^{(k)}) >$. This ensures no significant conflict between the historical pattern and the future condition. Finally, we get the counterfactual data $\tilde{\mathcal{D}} = \{(\mathbf{x}_h, \mathbf{c}_h, \varnothing, \tilde{\mathbf{c}}_f)_i\}_{i=1}^N$, where $\varnothing$ means the future sequence is absent. For training stability, the finetuning dataset combines factual data $\mathcal{D}$ and counterfactual data $\tilde{\mathcal{D}}$.

**Counterfactual Finetuning.** Since we lack the ground truth of future time series $\mathbf{x}_f$ for counterfactual data, we sample a Gaussian noise $\hat{\mathbf{x}}_{f,T} \sim \mathcal{N}(0, \mathbf{I})$ as the initial state, and gradually denoise it for diffusion training. For each denoising step $t$, we have noisy data $\hat{\mathbf{x}}_{f,t}$ to estimate the clear data $\hat{\mathbf{x}}_{f,0}$ through Eq. 3, and train the model to increase the DTTC scores with $\lambda_I, \lambda_E$ as loss weight:

$$\mathbf{I}_h, \mathbf{E}_h = E_{\text{ts}}(\mathbf{x}_h); \hat{\mathbf{I}}_f, \hat{\mathbf{E}}_f = E_{\text{ts}}(\hat{\mathbf{x}}_{f,0}); \tilde{\mathbf{E}}_f = E_{\text{text}}(\mathbf{c}_f),$$
$$\mathcal{L}_{\text{DTTC}}(\theta) = -\mathbb{E}\left(\lambda_I \cdot < \mathbf{I}_h, \hat{\mathbf{I}}_f > + \lambda_E \cdot < \hat{\mathbf{E}}_f, \tilde{\mathbf{E}}_f >\right). \tag{10}$$

By finetuning our model with counterfactual data, we enhance its adaptability to diverse future conditions. In contrast to prior methods (Melnychuk et al., 2022), which rely on ground-truth future sequences for training, our finetuning approach leverages semantic metrics to optimize the model without access to such ground truth. Importantly, finetuning on counterfactual data does not compromise forecasting performance on factual data, as demonstrated in Sec. 4.3.

# 4. Experiment

In this section, we describe the experimental setup and report results, structured around the following research questions (RQs): **RQ1**: Can TADIFF preserve the intrinsic features of the historical sequence while simultaneously satisfying the specified future conditions? **RQ2**: Can TADIFF adapt to diverse counterfactual conditions given the same historical sequence? **RQ3**: Does the DTTC metric enable providing reasonable and intuitive evaluations? The code are released at https://seqml.github.io/TADiff/.

## 4.1. Experiment Setup

**Datasets.** We utilize two categories of datasets: (i) Synthetic dataset. This includes **Synth** with human-constructed time series and textual descriptions, where each historical sequence is paired with diverse future conditions. Since the generation process of **Synth** dataset is fully under our control, ground truth future time series are available even in a counterfactual setting. (ii) Real-world dataset. This includes **Weather** (Xu et al., 2024), **ETTm1** (Zhou et al., 2021), **Exchange** (Lai et al., 2018), and **Traffic** (Leo, 2024), where we have the time series from real-world and textual descriptions from expert annotations or extrinsic tools. We further construct the counterfactual data through the method mentioned in Sec. 3.5 for evaluation. More details of dataset construction are provided in Appendix A.

**Evaluation Metrics.** We evaluate forecasts from two perspectives: (i) Numerical Accuracy. Mean absolute error (**MAE**) and mean squared error (**MSE**) are adopted when the ground truth of future is available; (ii) Semantic Consistency. Disentangled time series text consistency (**DTTC**) scores are adopted even when the ground truth is absent. As detailed in Sec. 3.4, **DTTC-I** measures consistency of forecasts with historical intrinsic features, **DTTC-E** measures consistency of forecasts with extrinsic future controls. Details of the DTTC model are provided in Appendix B.

**Baselines.** We compare against both unimodal and multimodal models. Unimodal baselines include deep learning models **Dlinear** (Zeng et al., 2023), **PatchTST** (Nie et al., 2022), and foundation model **Sundial** (Liu et al., 2025). Multimodal baselines include generation model **VerbalTS** (Gu et al., 2025) with text input, class-based counterfactual model **CT** (Melnychuk et al., 2022) with sequence and attribute inputs, text-based models **TimeMMD** (Liu et al., 2024b), **TimeCMA** (Liu et al., 2024a), and **IATSF** (Xu et al., 2024) with sequence and text inputs.

**Setting.** We adopt dataset-specific configurations for each dataset. All experiments are run on a single NVIDIA-A40 GPU with three random seeds, and we report both the mean and standard deviation. Details of training and finetuning are provided in Appendix D.

## 4.2. Main Results

We quantitatively evaluate the forecasting of TADIFF against baselines across all datasets, considering both factual and counterfactual settings. For the factual conditions

*Table 1.* Averaged forecasting performance under *factual* conditions. The best results are in **bold**, and the second-best results are underlined. Arrows ↑ (↓) indicate that higher (lower) is better.

| Datasets | Metric | DLinear | PatchTST | Sundial | VerbalTS | TimeCMA | TimeMMD | CT | IATSF | TADIFF (ours) |
|---|---|---|---|---|---|---|---|---|---|---|
| Synth | ↓ MAE | $0.73_{\pm 0.03}$ | $0.67_{\pm 0.00}$ | $0.76_{\pm 0.00}$ | $1.00_{\pm 0.00}$ | $0.82_{\pm 0.00}$ | $0.66_{\pm 0.00}$ | $0.57{\pm}0.00$ | $\underline{0.55}_{\pm 0.00}$ | $\mathbf{0.54}_{\pm 0.00}$ |
| | ↓ MSE | $0.87_{\pm 0.06}$ | $0.73_{\pm 0.01}$ | $1.01_{\pm 0.00}$ | $1.53_{\pm 0.01}$ | $1.09_{\pm 0.00}$ | $0.67_{\pm 0.00}$ | $\mathbf{0.34}_{\pm 0.00}$ | $\underline{0.48}_{\pm 0.01}$ | $0.51_{\pm 0.00}$ |
| | ↑ DTTC-I | $13.75_{\pm 0.18}$ | $\underline{14.65}_{\pm 0.06}$ | $13.73_{\pm 0.01}$ | $12.42_{\pm 0.03}$ | $13.64_{\pm 0.01}$ | $14.30_{\pm 0.02}$ | $13.43_{\pm 0.04}$ | $13.08_{\pm 0.01}$ | $\mathbf{15.37}_{\pm 0.03}$ |
| | ↑ DTTC-E | $64.68_{\pm 0.20}$ | $67.06_{\pm 0.05}$ | $64.25_{\pm 0.04}$ | $\underline{74.30}_{\pm 0.28}$ | $64.26_{\pm 0.05}$ | $68.87_{\pm 0.21}$ | $67.68_{\pm 0.21}$ | $63.32_{\pm 0.02}$ | $\mathbf{78.40}_{\pm 0.13}$ |
| ETTm1 | ↓ MAE | $0.84_{\pm 0.01}$ | $0.82_{\pm 0.02}$ | $0.90_{\pm 0.00}$ | $\underline{0.58}_{\pm 0.02}$ | $0.88_{\pm 0.00}$ | $0.79_{\pm 0.01}$ | $0.79_{\pm 0.02}$ | $0.82_{\pm 0.00}$ | $\mathbf{0.57}_{\pm 0.04}$ |
| | ↓ MSE | $1.52_{\pm 0.01}$ | $1.55_{\pm 0.05}$ | $1.90_{\pm 0.00}$ | $\underline{0.87}_{\pm 0.05}$ | $1.61_{\pm 0.01}$ | $1.37_{\pm 0.03}$ | $1.29_{\pm 0.05}$ | $1.42_{\pm 0.02}$ | $\mathbf{0.76}_{\pm 0.10}$ |
| | ↑ DTTC-I | $8.99_{\pm 0.39}$ | $\underline{9.97}_{\pm 0.12}$ | $9.96_{\pm 0.00}$ | $8.16_{\pm 0.04}$ | $8.34_{\pm 0.09}$ | $9.74_{\pm 0.19}$ | $8.44_{\pm 0.09}$ | $7.92_{\pm 0.04}$ | $\mathbf{10.01}_{\pm 0.03}$ |
| | ↑ DTTC-E | $86.33_{\pm 5.97}$ | $109.60_{\pm 1.63}$ | $96.98_{\pm 0.14}$ | $\underline{141.14}_{\pm 1.55}$ | $74.17_{\pm 1.01}$ | $108.14_{\pm 0.76}$ | $88.22_{\pm 1.47}$ | $80.65_{\pm 0.95}$ | $\mathbf{146.91}_{\pm 1.78}$ |
| Traffic | ↓ MAE | $0.52_{\pm 0.00}$ | $0.42_{\pm 0.02}$ | $\underline{0.40}_{\pm 0.00}$ | $0.80_{\pm 0.00}$ | $0.93_{\pm 0.00}$ | $0.41_{\pm 0.00}$ | $0.45_{\pm 0.00}$ | $0.91_{\pm 0.00}$ | $\mathbf{0.35}_{\pm 0.01}$ |
| | ↓ MSE | $0.43_{\pm 0.00}$ | $0.29_{\pm 0.02}$ | $0.34_{\pm 0.00}$ | $1.07_{\pm 0.01}$ | $1.07_{\pm 0.00}$ | $\underline{0.28}_{\pm 0.01}$ | $0.34_{\pm 0.01}$ | $1.05_{\pm 0.00}$ | $\mathbf{0.27}_{\pm 0.05}$ |
| | ↑ DTTC-I | $11.27_{\pm 0.07}$ | $12.97_{\pm 0.41}$ | $\underline{16.36}_{\pm 0.01}$ | $9.73_{\pm 0.05}$ | $6.31_{\pm 0.02}$ | $12.97_{\pm 0.36}$ | $11.12_{\pm 0.16}$ | $6.52_{\pm 0.02}$ | $\mathbf{17.90}_{\pm 0.07}$ |
| | ↑ DTTC-E | $48.31_{\pm 0.14}$ | $58.32_{\pm 2.02}$ | $66.14_{\pm 0.09}$ | $53.75_{\pm 0.30}$ | $21.71_{\pm 0.12}$ | $\underline{58.86}_{\pm 1.42}$ | $54.10_{\pm 1.49}$ | $24.21_{\pm 0.22}$ | $\mathbf{81.38}_{\pm 0.15}$ |
| Exchange | ↓ MAE | $0.15_{\pm 0.00}$ | $\underline{0.14}_{\pm 0.00}$ | $\underline{0.14}_{\pm 0.00}$ | $0.21_{\pm 0.00}$ | $0.15_{\pm 0.00}$ | $0.22_{\pm 0.00}$ | $0.23_{\pm 0.01}$ | $\mathbf{0.12}_{\pm 0.00}$ | $0.15_{\pm 0.01}$ |
| | ↓ MSE | $\mathbf{0.04}_{\pm 0.00}$ | $\mathbf{0.04}_{\pm 0.00}$ | $\mathbf{0.04}_{\pm 0.00}$ | $0.10_{\pm 0.00}$ | $\mathbf{0.04}_{\pm 0.00}$ | $\underline{0.08}_{\pm 0.00}$ | $0.10_{\pm 0.01}$ | $\mathbf{0.04}_{\pm 0.00}$ | $\mathbf{0.04}_{\pm 0.00}$ |
| | ↑ DTTC-I | $16.62_{\pm 0.02}$ | $16.78_{\pm 0.17}$ | $\underline{17.04}_{\pm 0.02}$ | $15.62_{\pm 0.03}$ | $16.62_{\pm 0.02}$ | $16.03_{\pm 0.03}$ | $15.71_{\pm 0.02}$ | $16.33_{\pm 0.03}$ | $\mathbf{17.37}_{\pm 0.04}$ |
| | ↑ DTTC-E | $204.90_{\pm 0.17}$ | $208.68_{\pm 0.67}$ | $206.49_{\pm 0.31}$ | $\underline{212.13}_{\pm 0.12}$ | $204.64_{\pm 0.10}$ | $184.25_{\pm 1.58}$ | $199.23_{\pm 1.38}$ | $209.12_{\pm 0.33}$ | $\mathbf{216.03}_{\pm 3.11}$ |
| Weather | ↓ MAE | $0.26_{\pm 0.00}$ | $\underline{0.19}_{\pm 0.00}$ | $0.24_{\pm 0.00}$ | $0.51_{\pm 0.02}$ | $0.30_{\pm 0.00}$ | $0.27_{\pm 0.00}$ | $0.31_{\pm 0.05}$ | $\mathbf{0.18}_{\pm 0.00}$ | $\mathbf{0.18}_{\pm 0.00}$ |
| | ↓ MSE | $0.15_{\pm 0.00}$ | $\underline{0.11}_{\pm 0.00}$ | $0.14_{\pm 0.00}$ | $0.46_{\pm 0.03}$ | $0.19_{\pm 0.00}$ | $0.13_{\pm 0.00}$ | $0.17_{\pm 0.04}$ | $\mathbf{0.09}_{\pm 0.00}$ | $\underline{0.11}_{\pm 0.00}$ |
| | ↑ DTTC-I | $26.79_{\pm 0.00}$ | $\underline{26.83}_{\pm 0.02}$ | $\underline{26.83}_{\pm 0.01}$ | $24.85_{\pm 0.06}$ | $\mathbf{26.84}_{\pm 0.00}$ | $26.41_{\pm 0.02}$ | $26.24_{\pm 0.32}$ | $26.54_{\pm 0.00}$ | $26.65_{\pm 0.04}$ |
| | ↑ DTTC-E | $29.60_{\pm 0.02}$ | $30.25_{\pm 0.04}$ | $29.44_{\pm 0.01}$ | $31.23_{\pm 0.02}$ | $29.45_{\pm 0.01}$ | $29.25_{\pm 0.08}$ | $29.98_{\pm 0.27}$ | $29.81_{\pm 0.00}$ | $\mathbf{31.55}_{\pm 0.15}$ |

*Table 2.* Averaged forecasting performance under *counterfactual* conditions. The best results are in **bold**, and the second-best results are underlined. Arrows ↑ (↓) indicate that higher (lower) is better.

| Datasets | Metric | DLinear | PatchTST | Sundial | VerbalTS | TimeCMA | TimeMMD | CT | IATSF | TADIFF (ours) |
|---|---|---|---|---|---|---|---|---|---|---|
| ETTm1 | ↑ DTTC-I | $8.96_{\pm 0.40}$ | $\underline{10.51}_{\pm 0.11}$ | $9.94_{\pm 0.00}$ | $8.62_{\pm 0.04}$ | $9.47_{\pm 0.61}$ | $10.14_{\pm 0.06}$ | $8.43_{\pm 0.10}$ | $7.93_{\pm 0.04}$ | $\mathbf{10.59}_{\pm 0.02}$ |
| | ↑ DTTC-E | $96.45_{\pm 4.89}$ | $113.02_{\pm 1.39}$ | $107.14_{\pm 0.02}$ | $\underline{140.95}_{\pm 1.57}$ | $103.42_{\pm 7.89}$ | $113.65_{\pm 0.49}$ | $93.24_{\pm 2.09}$ | $87.71_{\pm 0.62}$ | $\mathbf{148.72}_{\pm 1.21}$ |
| Traffic | ↑ DTTC-I | $11.33_{\pm 0.08}$ | $12.99_{\pm 0.39}$ | $\underline{16.32}_{\pm 0.01}$ | $9.44_{\pm 0.05}$ | $6.36_{\pm 0.03}$ | $12.96_{\pm 0.43}$ | $11.15_{\pm 0.14}$ | $6.57_{\pm 0.08}$ | $\mathbf{18.08}_{\pm 0.11}$ |
| | ↑ DTTC-E | $48.41_{\pm 0.18}$ | $54.86_{\pm 1.70}$ | $\underline{64.79}_{\pm 0.04}$ | $53.69_{\pm 0.32}$ | $22.71_{\pm 0.03}$ | $56.10_{\pm 1.01}$ | $52.67_{\pm 1.55}$ | $24.62_{\pm 0.24}$ | $\mathbf{78.58}_{\pm 0.06}$ |
| Exchange | ↑ DTTC-I | $16.61_{\pm 0.02}$ | $16.79_{\pm 0.17}$ | $\underline{17.03}_{\pm 0.01}$ | $15.29_{\pm 0.06}$ | $16.79_{\pm 0.08}$ | $16.25_{\pm 0.17}$ | $15.67_{\pm 0.04}$ | $16.31_{\pm 0.05}$ | $\mathbf{17.30}_{\pm 0.04}$ |
| | ↑ DTTC-E | $200.45_{\pm 0.04}$ | $201.30_{\pm 1.08}$ | $200.57_{\pm 0.13}$ | $\underline{211.37}_{\pm 0.61}$ | $201.84_{\pm 0.39}$ | $196.16_{\pm 2.10}$ | $194.49_{\pm 0.85}$ | $199.65_{\pm 0.26}$ | $\mathbf{213.94}_{\pm 2.18}$ |
| Weather | ↑ DTTC-I | $26.79_{\pm 0.01}$ | $26.83_{\pm 0.02}$ | $26.83_{\pm 0.00}$ | $24.43_{\pm 0.05}$ | $\underline{26.84}_{\pm 0.00}$ | $26.75_{\pm 0.11}$ | $26.25_{\pm 0.29}$ | $26.53_{\pm 0.01}$ | $\mathbf{26.93}_{\pm 0.04}$ |
| | ↑ DTTC-E | $29.04_{\pm 0.03}$ | $29.25_{\pm 0.05}$ | $28.91_{\pm 0.01}$ | $\underline{30.16}_{\pm 0.05}$ | $29.00_{\pm 0.01}$ | $29.30_{\pm 0.18}$ | $29.23_{\pm 0.18}$ | $29.18_{\pm 0.01}$ | $\mathbf{30.99}_{\pm 0.14}$ |

(Tab. 1), where ground-truth of the future sequences is available, we report MAE, MSE, DTTC-I, and DTTC-E. For the counterfactual conditions (Tab. 2), where the ground-truth of the future time series is absent, we only report DTTC-I and DTTC-E. All experiments are run three times with different random seeds, and we report the mean and standard deviation of each metric. We further provide the implementation settings, case study, model efficiency analysis, and other extended analysis in Appendix E.

**Finding 1:** TADIFF *achieves superior numerical accuracy and semantic consistency on both synthetic and real datasets.* As shown in Tab. 1, TADIFF consistently outperforms baselines across multiple datasets, delivering the best performance in both numerical accuracy and semantic consistency on Synth, ETTm1, Traffic, and Exchange, and achieving the best MSE and DTTC-E on Weather. The generation baselines fail to account for the impact of historical sequences. While they achieve high semantic consistency with future conditions, they struggle with numerical accuracy and alignment with historical patterns. The forecasting baselines neglect multimodal information or the combined influence of history and future, resulting in suboptimal performance. This addresses **RQ1** by demonstrating that TAD-IFF better balances the influence of historical sequences and future conditions, leading to more accurate forecasts that remain semantically consistent with both constraints.

**Finding 2:** TADIFF *exhibits strong adaptability and generalization for forecasting under diverse future conditions.* We evaluated TADIFF on the test splits of the synthetic and real-world data in the counterfactual setting, which include diverse and unseen condition combinations. The results of synthetic data in Tab. 1 and real-world data in Tab. 2 show that TADIFF achieves consistently better semantic consistency when forecasting under diverse counterfactual conditions. This addresses **RQ2**, demonstrating that TAD-IFF possesses stronger generalization and adaptability when conditioned on varying future scenarios.

### 4.3. Ablation Study and Extended Analysis

**Finding 3:** *Finetuning with counterfactual data preserves numerical accuracy while enhancing semantic consistency.*

*Table 3.* The ablation study of CF and TA on **ETTm1** and **Traffic** datasets. CF represents the fine-tuning on the counterfactual data. TA represents the text-attribution mechanism.

| Datasets | ETTm1 | | | | | | Traffic | | | | | |
|---|---|---|---|---|---|---|---|---|---|---|---|---|
| Setting | Factual | | | | Counterfactual | | Factual | | | | Counterfactual | |
| Metric | ↓MAE | ↓MSE | ↑DTTC-I | ↑DTTC-E | ↑DTTC-I | ↑DTTC-E | ↓MAE | ↓MSE | ↑DTTC-I | ↑DTTC-E | ↑DTTC-I | ↑DTTC-E |
| TADIFF | **0.57**$_{\pm 0.04}$ | **0.76**$_{\pm 0.10}$ | **10.01**$_{\pm 0.03}$ | **146.91**$_{\pm 1.78}$ | **10.59**$_{\pm 0.02}$ | **148.72**$_{\pm 1.21}$ | 0.35$_{\pm 0.03}$ | 0.27$_{\pm 0.05}$ | **17.90**$_{\pm 0.07}$ | **81.38**$_{\pm 0.15}$ | **18.08**$_{\pm 0.11}$ | **78.58**$_{\pm 0.06}$ |
| w/o CF | 0.58$_{\pm 0.05}$ | 0.82$_{\pm 0.17}$ | 9.48$_{\pm 0.06}$ | 144.74$_{\pm 1.65}$ | 10.06$_{\pm 0.04}$ | 146.73$_{\pm 1.12}$ | **0.31**$_{\pm 0.01}$ | **0.24**$_{\pm 0.01}$ | 17.57$_{\pm 0.03}$ | 80.84$_{\pm 0.21}$ | 17.31$_{\pm 0.03}$ | 78.11$_{\pm 0.22}$ |
| w/o TA | 0.59$_{\pm 0.04}$ | 0.77$_{\pm 0.04}$ | 9.05$_{\pm 0.04}$ | 142.36$_{\pm 1.50}$ | 9.35$_{\pm 0.07}$ | 141.01$_{\pm 1.75}$ | 0.49$_{\pm 0.01}$ | 0.78$_{\pm 0.16}$ | 15.41$_{\pm 0.11}$ | 76.34$_{\pm 0.22}$ | 15.28$_{\pm 0.04}$ | 73.52$_{\pm 0.25}$ |

As shown in Tab. 3, we compare the forecasting performance before and after fine-tuning on the counterfactual data. We observe that fine-tuning improves the model's adaptability and leads to higher semantic consistency (DTTC-I and DTTC-E) in both factual and counterfactual forecasting. Although the fine-tuning process is primarily designed to enhance semantic consistency metrics, the model consistently maintains strong numerical accuracy (MAE and MSE). These results further demonstrate that the semantic consistency metrics do not conflict with the numerical accuracy metrics in time series forecasting.

**Finding 4:** *Text-attribution substantially improves the consistency of forecasts with both historical sequence and future conditions.* As demonstrated in Tab. 3, we compare the forecasting performance with and without text-attribution. The results show that text-attribution improves both semantic consistency (DTTC-I and DTTC-E) and numerical accuracy (MAE and MSE), with particularly notable gains in DTTC-I and DTTC-E under the counterfactual setting. This indicates that attribution effectively balances the influence of historical patterns and future conditions, especially when there are conflicts between them. We further prove the text-attribution can decouple the intrinsic features of sequences from extrinsic conditions in Appendix E.3.

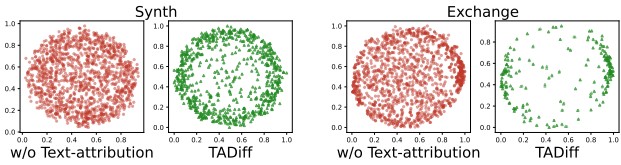

*Figure 4.* The initial noise distribution between the TADIFF (green) and TADIFF without text-attribution (red) on **Synth** (left) and **Exchange** (right) datasets.

**Finding 5:** *Text-attribution optimizes the initial noise space of the diffusion model.* We compare the initial noise distributions of TADIFF with and without text-attribution using t-SNE (Van der Maaten & Hinton, 2008) in Fig. 4. The results show that the initial noise distribution with text-attribution forms more compact clusters compared to the random Gaussian distribution in the attribution-free setting. This indicates that text-attribution helps preserve the intrinsic features of the historical sequence, thereby enabling a more effective conditional denoising process and ultimately producing more accurate forecasts, as discussed in Sec. 3.2.

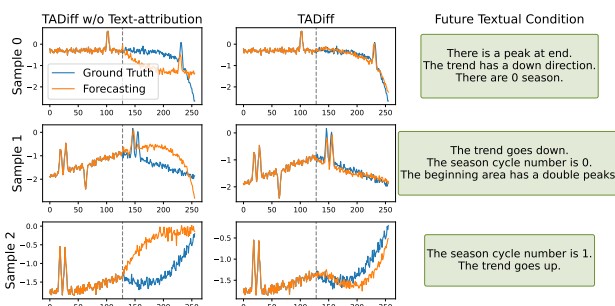

*Figure 5.* Qualitative comparison of forecasting on Synth dataset. Column 1: the forecasting of TADiff without text-attribution; Column 2: the forecasting of TADiff with text-attribution; Column 3: the future textual condition.

**Finding 6:** *Text-attribution facilitates the capture of intrinsic features in history, enhancing the incorporation of future textual influences.* As illustrated in Fig. 5, we qualitatively compare forecasts with and without text-attribution on the Synth dataset. The generation process of the Synth dataset guarantees that the historical and future time series share the same trend type, as detailed in Appendix A.2.2. The visualizations show that with text-attribution, TADiff more effectively captures the invariant trend type of historical sequence and aligns more closely with the semantics of future text, leading to more accurate and reasonable forecasts.

*Table 4.* Evaluation of DTTC models on different datasets. Given a future sequence, DTTC-I and DTTC-E report the accuracy of retrieving the most similar historical sequence and future condition, respectively, from three candidates.

| Dataset | Synth | ETTm1 | Traffic | Exchange | Weather |
|---|---|---|---|---|---|
| DTTC-I | 84.57% | 55.06% | 91.90% | 84.60% | 79.96% |
| DTTC-E | 96.36% | 95.93% | 97.69% | 98.44% | 77.91% |

**Finding 7:** *DTTC model effectively captures the intrinsic features of time series and the semantics of extrinsic conditions.* We evaluate DTTC using a retrieval-based protocol, following contrastive learning practices (Radford et al., 2021). The DTTC model is trained on the training split of factual data for each dataset and evaluated on the test split. Given a future time series $\mathbf{x}_f^{(1)}$, the DTTC model retrieves the paired historical sequence $\mathbf{x}_h^{(1)}$ from a candidate set containing two additional randomly sampled sequences

$\{\mathbf{x}_h^{(1)}, \mathbf{x}_h^{(2)}, \mathbf{x}_h^{(3)}\}$. Similarly, we retrieve the paired textual condition $\mathbf{c}_f^{(1)}$ from a set augmented with two random alternatives $\{\mathbf{c}_f^{(1)}, \mathbf{c}_f^{(2)}, \mathbf{c}_f^{(3)}\}$. As shown in Tab. 4, both DTTC-I and DTTC-E achieve high retrieval accuracy, demonstrating that DTTC effectively aligns the intrinsic features of historical and future series, as well as the semantics between future series and extrinsic conditions. This supports the reliability of the proposed metrics and addresses **RQ3**.

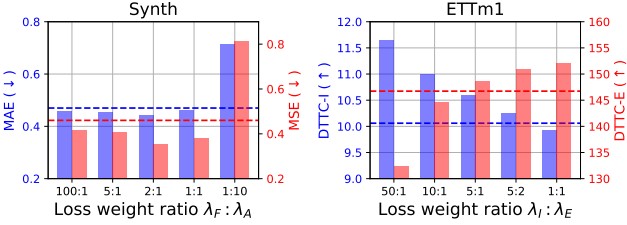

*Figure 6.* Sensitivity study on loss weight ratio. In the left figure, we study the impact of $\lambda_F : \lambda_A$ in joint training, where the dotted lines indicate the performance of TADIFF without attribution. In the right figure, we study the impact of $\lambda_I : \lambda_E$ in counterfactual finetuning, where the dotted lines indicate the performance of TADIFF without finetuning.

**Finding 8:** TADIFF *is robust to the training hyperparameters.* We investigate the effect of the loss weight ratio during training and counterfactual finetuning, and the results are shown in Fig. 6. The influence of $\lambda_F : \lambda_A$ remains stable in most cases, and significant performance differences are observed only when the ratios take extreme values. In counterfactual fine-tuning, a trade-off arises between DTTC-I and DTTC-E, which is intuitive and expected, since these two metrics are directly linked to their corresponding optimization objectives during fine-tuning. We adopt a balanced loss weight ratio to achieve simultaneous improvements in both metrics compared with forecasting without finetuning.

## 5. Conclusion

In this work, we introduce counterfactual time series forecasting with textual conditions, aiming to explore how time series evolve under complex, stochastic, and counterfactual future conditions. Existing methods fail to account for conflicts between historical patterns and future conditions, and they struggle with evaluation when ground truth is unavailable in counterfactual settings. To address these challenges, we propose TADIFF, a text-attributive diffusion model trained on factual data and fine-tuned with counterfactual data. For evaluation, we design metrics based on the DTTC model to assess the semantic consistency of forecasts. Experiments demonstrate TADIFF's superiority in counterfactual forecasting and the reliability of our evaluation framework. While there are still limitations, particularly in evaluating forecasts of varying lengths with model-based metrics, our work opens new avenues for modeling and as-

sessing counterfactual forecasts under complex conditions, with potential benefits for real-world decision-making.

## Impact Statement

This work introduces a novel task of counterfactual time series forecasting under textual conditions. We anticipate several positive societal impacts: (i) assisting humans in forecasting future events and making decisions under complex, stochastic assumptions; (ii) offering innovative and reliable evaluation methods for time series forecasting; and (iii) inspiring approaches of better leveraging multimodal information in time series forecasting.

While we do not foresee immediate or direct negative societal consequences stemming from this work, we recognize that, like other generative technologies, it may be susceptible to misuse. Therefore, responsible use, ethical oversight, and ongoing monitoring are crucial to ensure that the technology is applied for the greater good.

## Acknowledgment

The research was mainly supported by National Natural Science Foundation of China (Grant No. 62406193). And we are also supported by ShanghaiTech AI Initiative (Grant No. AI2026B08). The authors gratefully acknowledge further assistance provided by the Shanghai Frontiers Science Center of Human-centered Artificial Intelligence, the MoE Key Lab of Intelligent Perception and Human-Machine Collaboration, the ShanghaiTech GenAI Platform, and the HPC Platform of ShanghaiTech University.

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

# A. Datasets

Our experimental framework is built upon two primary categories of datasets: **synthetic** and **real-world**. The real-world category is further subdivided into datasets that are originally unimodal (time series only) and those that are multimodal (containing real-world captions). Each of these types requires a distinct data processing pipeline.

For the **unimodal real-world datasets** (ETTm1, Traffic, Exchange), we adopt the augmentation paradigm introduced by VerbalTS (Gu et al., 2025). This involves a two-stage process: first, we extract structured attributes from the raw time series data, then these attributes are used to generate corresponding textual descriptions.

For the **synthetic dataset** (Synth), the attributes are not extracted but are algorithmically predefined at the start of the generation process. In contrast, since the **multimodal real-world dataset** (Weather) already contains naturally occurring time series-text pairs, our task is to extract structured attributes directly from the provided real-world captions. Finally, for the synthetic and multimodal real-world datasets, the textual descriptions are generated from their respective attributes by populating predefined prompt templates.

The specific implementation details for each of these processes are provided in the corresponding dataset sections that follow.

## A.1. Forecasting Task Formulation

Based on these processed datasets, we formulate two distinct forecasting settings, **Factual** and **Counterfactual**, to evaluate model performance under different types of conditions. The implementation of these settings varies depending on the nature of the dataset. We define two types of forecasting settings as follows:

**Factual Forecasting.** This task is consistent across all dataset types and is represented by the tuple $(\mathbf{x}_h, \mathbf{c}_h, \mathbf{x}_f, \mathbf{c}_f)$. The objective is to predict the actual future time series $\mathbf{x}_f$ given the historical context $(\mathbf{x}_h, \mathbf{c}_h)$ and the corresponding true, observed future condition $\mathbf{c}_f$.

**Counterfactual Forecasting.** This task evaluates the model's ability to forecast under a future condition that deviates from the observed or expected trajectory. Its implementation differs between real-world and synthetic data:

- For Synthetic Datasets, the task retains the tuple structure $(\mathbf{x}_h, \mathbf{c}_h, \mathbf{x}_f, \mathbf{c}_f)$. Because we control the data generation process, we can still synthesize a corresponding ground-truth future time series $\mathbf{x}_f$ that matches this counterfactual condition.

- For Real-World Datasets, the task is represented by the tuple $(\mathbf{x}_h, \mathbf{c}_h, \varnothing, \tilde{\mathbf{c}}_f)$. Here, $\tilde{\mathbf{c}}_f$ is a hypothetical future condition that the combination of $\mathbf{x}_h$ and $\tilde{\mathbf{c}}_f$ did not actually occur in the real world. Consequently, a ground-truth future time series for this scenario does not exist and is represented by the empty set ($\varnothing$), as the outcome is inherently unobserved.

## A.2. Synthetic Dataset

In this work, we utilize a dataset we term Synth. The generation process is attribute-driven, where predefined attributes govern the synthesis of both time series and their textual descriptions.

### A.2.1. ATTRIBUTE FRAMEWORK FOR SYNTHETIC GENERATION

The generation of each synthetic time series is governed by six attributes. These attributes are randomly sampled for each instance to control the series' fundamental structure, localized events, and stochastic properties, ensuring a diverse data distribution. These attributes are defined as follows:

- **Trend Type:** We define four distinct trend types to model different long-term behaviors: *linear*, *quadratic*, *exponential*, and *logistic*. The base trend is generated from mathematical formulas, with some types (e.g., logistic) being normalized to the range $[-1, 1]$ for training stability.

- **Trend Direction:** Each generated trend is assigned one of two directions: *upward* or *downward*. This is implemented as a simple sign multiplier ($+1$ or $-1$) on the base trend component.

- **Seasonality Cycles:** To introduce periodic patterns, each series is assigned a specific number of cycles, with the count selected from $\{0, 1, 2, 4\}$. This component is modeled as a sinusoidal wave, $\mathbf{x}_{season} = a\sin(2\pi t + \phi)$, where the amplitude $a \sim \mathcal{U}(0.4, 0.6)$ and phase $\phi \sim \mathcal{U}(0, 2\pi)$ are randomly sampled to ensure diversity.

- **Local Shapelets:** We introduce one of five predefined local shapelets: *nothing, peak, sag, double peaks,* or *platform*. Each time series is divided into three equal segments (beginning, middle, end). Within each segment, a shapelet is added with a specific probability.

- **Noise:** We inject additive Gaussian noise into each sample. The noise is sampled independently for each time step from a zero-mean Gaussian distribution, $\mathbf{x}_{noise} \sim \mathcal{N}(0, \sigma^2)$, where the standard deviation $\sigma$ is itself randomly sampled from a small range to ensure variability in the noise level across the dataset.

- **Bias:** A constant bias is added to each time series to vary its global vertical offset. The bias value is sampled from one of three predefined ranges (negative, zero, or positive), ensuring variability in the series' mean value across the dataset.

The six attributes are divided into intrinsic features and extrinsic conditions, where **Trend Type**, **Noise**, and **Bias** belong to intrinsic features; **Trend Direction**, **Seasonality Cycles**, and **Local Shapelets** belong to extrinsic conditions. The intrinsic features will remain unchanged between the historical and future sequences, and only the extrinsic conditions will be described in the texts.

### A.2.2. SYNTH DATASET

The `Synth` dataset is algorithmically generated by randomly sampling attributes for each instance, ensuring high variability. The entire process yields 14,000 unique samples, which are then partitioned into training (11,200 samples), validation (1,400), and test (1,400) sets using an 8:1:1 ratio.

**Attribute Generation.**  Each time series sample is governed by the attributes proposed above that define its structure. For each sample, we generate a related pair of historical and future segments. A key constraint is that **Trend Type**, **Noise**, and **Bias** are shared between the history and the future, as they represent the invariant intrinsic features of the time series. In contrast, the remaining attributes, **Trend Direction**, **Seasonality Cycles**, and **Local Shapelets**, are extrinsic conditions that may vary between the history and the future and are intended to be described through textual captions.

**Caption Generation.**  A corresponding textual caption is generated for both the historical and future segments using randomized prompt templates. This process ensures diversity while maintaining semantic consistency with the attributes. A representative example of a generated caption pair is as follows:

- **History Caption:** "The trend goes up. There is a sag at end. There is 1 season."

- **Future Caption:** "There is a sag at end. The season cycle number is 0. The trend has a down direction."

This example illustrates a scenario where the trend direction reverses and the number of seasonal cycles changes, providing a challenging test case for conditional forecasting.

### A.3. Real-world Datasets

Real-world datasets consist of observational data from real-world processes. Depending on whether the original dataset contains pre-existing text, we categorize them as unimodal or multimodal and apply a distinct data preparation pipeline to each.

### A.3.1. UNIMODAL DATASETS

Unimodal datasets in our study initially contain only time series data. To prepare them for multimodal inputs, we follow the augmentation paradigm introduced by VerbalTS (Gu et al., 2025). First, we perform **attribute extraction** by applying the `tsfresh` library (Christ et al., 2018) to the raw time series to derive a set of structured statistical features. These extracted features, which are categorized into global and local characteristics, are summarized in Tab. 5. Second, we perform **caption generation** by using these attributes to populate predefined prompt templates, thereby creating a synthetic textual description for each time series sample. The specific datasets processed with this methodology are as follows.

*Table 5.* Summary of global and local features extracted for data augmentation.

|  | Feature Name | Description |
|---|---|---|
| **Global Features** | Skewness | The asymmetry of the distribution. |
|  | Kurtosis | The sharpness of the distribution. |
|  | Linear Trend | The overall trend direction and rate of change. |
|  | FFT Frequency | The dominant periodicity in frequency domain. |
| **Local Features** | Local Linear Trend | The trend direction within each segment. |
|  | Number of Peaks | The number of local maxima within each segment. |

**ETTm1.** The `ETTm1` dataset (Zhou et al., 2021) contains 7 variables of electricity transformer data sampled every 15 minutes. We partitioned the data and employed a sliding window (history 48, stride 48, horizon 48) to generate 8,120 training, 1,008 validation, and 1,008 test samples.

**Traffic.** The `Traffic` dataset (Leo, 2024) contains 3 variables of traffic data from Istanbul. We downsampled the original 1-minute data to an hourly frequency. The data was split and processed with a sliding window (history 48, stride 4, horizon 48) to create 8,106 training, 951 validation, and 951 test samples.

**Exchange.** The `Exchange` dataset (Lai et al., 2018) consists of 8 daily exchange rates. We used a sliding window (history 48, stride 12, horizon 48) to generate 3,984 training, 448 validation, and 448 test samples.

### A.3.2. MULTIMODAL DATASET

Multimodal datasets are those that natively contain both time series data and textual descriptions. Instead of generating text from features, we perform **attribute extraction** from the existing texts. This allows us to generate a new, standardized textual description.

**Weather.** The `Weather` dataset (Xu et al., 2024) from the Max Planck Institute contains 3 atmospheric variables sampled every 10 minutes. We used data from 2014-2022, splitting it into training, validation, and test sets. A sliding window (history 36, stride 36, horizon 36) was used for sample generation. The training, validation and test samples are 30,573, 4,377 and 4,341. The accompanying textual descriptions were processed using the methodology mentioned before. Specifically, we employed GPT-4 (Achiam et al., 2023) to parse each caption and identify values for a predefined set of seven attributes:

- **Season**: spring, summer, fall, winter.

- **Time of Day**: early morning, morning, afternoon, evening.

- **Weather Condition**: sunny, cloudy, rain, foggy, snowy.

- **Temperature Trend**: increase, decrease, steady.

- **Wind Direction**: S, N, W, E, SW, SE, NW, NE.

- **Atmospheric Condition**: low, average, high.

- **Humidity Level**: low, average, high.

An "unknown" category was also included for cases where information was not present in the text.

## B. DTTC Model

As introduced in Sec. 3.4, the DTTC (Disentangled Time Series and Text Consistency) model is designed to evaluate forecasts by measuring their consistency between both historical patterns and future textual conditions. The DTTC model leverages contrastive learning to disentangle time series representations into intrinsic features and extrinsic condition-dependent attributes. This section will introduce the architecture and training of the DTTC model in detail.

Conceptually similar to the CLIP model (Radford et al., 2021), the DTTC framework is composed of a dual-encoder architecture: a time series encoder $E_{\text{ts}}$ and a text encoder $E_{\text{text}}$, where the complete parameters can be expressed as $\phi$. During training, the model processes tuples of data, each containing a historical sequence $\mathbf{x}_h$, its textual conditions $\mathbf{c}_h$, a corresponding future forecast $\mathbf{x}_f$, and the future textual conditions $\mathbf{c}_f$.

The time series encoder, which utilizes the PatchTST architecture (Nie et al., 2022), is trained to disentangle an input time series $\mathbf{x}$ into two distinct representations: an intrinsic feature vector $\mathbf{I}$ and an extrinsic feature vector $\mathbf{E}$. It processes both the historical and future sequences:

- For historical sequence $\mathbf{x}_h$, the output is $(\mathbf{I}_h, \mathbf{E}_h) = E_{\text{ts}}(\mathbf{x}_h)$.

- For future forecast $\mathbf{x}_f$, the output is $(\mathbf{I}_f, \mathbf{E}_f) = E_{\text{ts}}(\mathbf{x}_f)$.

Correspondingly, the text encoder is designed to map the textual conditions, $\mathbf{c}$, into the same extrinsic feature space, producing an embedding $\tilde{\mathbf{E}}$. In order to handle long text inputs, we adopt the tokenizer from the pre-trained Long-CLIP (Zhang et al., 2024) model. The parameters of the encoder itself, however, are trained from scratch. This encoder also processes both historical and future conditions:

- For historical conditions $\mathbf{c}_h$, the output is $\tilde{\mathbf{E}}_h = E_{\text{text}}(\mathbf{c}_h)$.

- For future conditions $\mathbf{c}_f$, the output is $\tilde{\mathbf{E}}_f = E_{\text{text}}(\mathbf{c}_f)$.

The training process is guided by two distinct contrastive learning objectives: an intrinsic consistency loss and an extrinsic consistency loss.

The intrinsic consistency loss, $\mathcal{L}_I$, enforces that the intrinsic features of the historical sequence $\mathbf{I}_h$ and the future forecast $\mathbf{I}_f$ remain aligned. This encourages the model to preserve the fundamental, time-invariant dynamics of the series across the historical and future segments. The loss is defined as:

$$\mathcal{L}_I(\phi) = -\frac{1}{B} \sum_{i=1}^{B} \log \frac{\exp(<\mathbf{I}_h[i], \mathbf{I}_f[i]>)}{\sum_{j=1}^{B} \exp(<\mathbf{I}_h[i], \mathbf{I}_f[j]>)}, \tag{11}$$

The second objective, the extrinsic consistency loss, $\mathcal{L}_E$, ensures that the extrinsic features extracted from the time series align with the features from their corresponding textual conditions. This loss comprising two terms that respectively align historical features ($\mathbf{E}_h$ and $\tilde{\mathbf{E}}_h$) and future features ($\mathbf{E}_f$ and $\tilde{\mathbf{E}}_f$):

$$\mathcal{L}_E(\phi) = -\frac{1}{2B} \sum_{i=1}^{B} \left( \log \frac{\exp(<\mathbf{E}_h[i], \tilde{\mathbf{E}}_h[i]>)}{\sum_{j=1}^{B} \exp(<\mathbf{E}_h[i], \tilde{\mathbf{E}}_h[j]>)} + \log \frac{\exp(<\mathbf{E}_f[i], \tilde{\mathbf{E}}_f[i]>)}{\sum_{j=1}^{B} \exp(<\mathbf{E}_f[i], \tilde{\mathbf{E}}_f[j]>)} \right), \tag{12}$$

where $< \cdot, \cdot >$ denotes the inner product and $B$ is the batch size. By jointly optimizing these two objectives, the model learns a representation space suitable for evaluating forecast consistency. The pseudocode for the DTTC model is presented in Algorithm 1 and the detailed model architecture is in Fig. 7.

## C. Model Architecture

The architecture of the noise estimator of TADIFF, illustrated in Fig. 8, is inspired by the DiT model (Peebles & Xie, 2023), and is adapted for the specific demands of time series forecasting with multi-modal conditioning.

First, the inputs are converted into embeddings. The historical time series $\mathbf{x}_{h,0} \in \mathbb{R}^{L_h}$ and the noisy future time series $\mathbf{x}_{f,t} \in \mathbb{R}^{L_f}$ are separately passed through a patchify module to obtain token sequences $\mathbf{P}_h \in \mathbb{R}^{N_h \times D}$ and $\mathbf{P}_f \in \mathbb{R}^{N_f \times D}$, which are then concatenated along the token dimension to form $\mathbf{P} \in \mathbb{R}^{N \times D}$ with $N = N_h + N_f$. Meanwhile, the diffusion step $t$ is transformed into a time embedding $\mathbf{e}_{\text{time}} \in \mathbb{R}^D$, and the textual condition $\mathbf{c}_f$ is processed by a text encoder to yield the text embedding $\mathbf{e}_{\text{text}} \in \mathbb{R}^D$. The text encoder is composed of a pre-trained Long-CLIP tokenizer (Zhang et al., 2024) and two transformer layers trained from scratch. We combine these two embeddings by element-wise addition to obtain the final condition embedding $\mathbf{e}_c = \mathbf{e}_{\text{time}} + \mathbf{e}_{\text{text}} \in \mathbb{R}^D$.

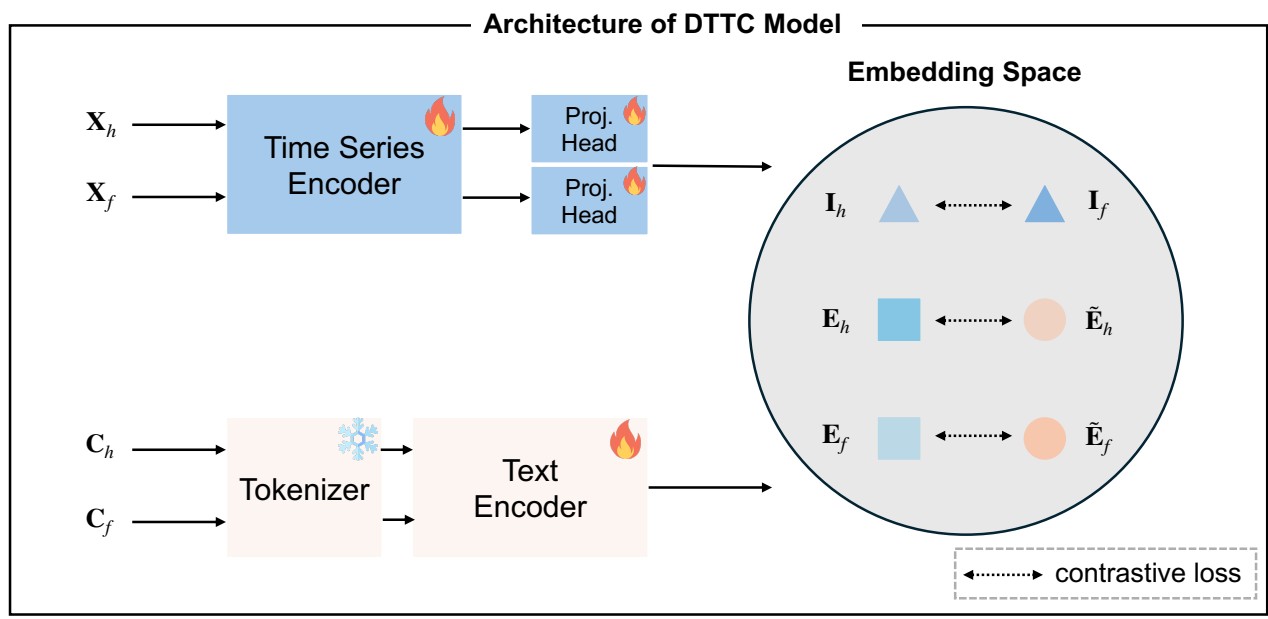

*Figure 7.* The model architecture of the DTTC Model.

---

**Algorithm 1** Pseudocode for the DTTC Model Training

---

**Input:** A batch of training tuples $(\mathbf{x}_h^{(i)}, \mathbf{c}_h^{(i)}, \mathbf{x}_f^{(i)}, \mathbf{c}_f^{(i)})_{i=1}^B$, where $\mathbf{x}$ are time series and $\mathbf{c}$ are texts.
**Output:** Total loss $\mathcal{L}_{\text{total}}$.

1: **# Disentangle features using encoders**
2: $(\mathbf{I}_h, \mathbf{E}_h) \leftarrow E_{\text{ts}}(\mathbf{x}_h)$            ▷ Disentangled historical features.
3: $(\mathbf{I}_f, \mathbf{E}_f) \leftarrow E_{\text{ts}}(\mathbf{x}_f)$            ▷ Disentangled future features.
4: $\tilde{\mathbf{E}}_h \leftarrow E_{\text{text}}(\mathbf{C}_h)$            ▷ Historical text features.
5: $\tilde{\mathbf{E}}_f \leftarrow E_{\text{text}}(\mathbf{C}_f)$            ▷ Future text features.

6: **# Compute pairwise similarity matrices**
7: $\mathbf{S}_I \leftarrow \text{Sim}(\mathbf{I}_h, \mathbf{I}_f)$            ▷ $\mathbf{S}_I \in \mathbb{R}^{B \times B}$, intrinsic similarity matrix.
8: $\mathbf{S}_{Eh} \leftarrow \text{Sim}(\mathbf{E}_h, \tilde{\mathbf{E}}_h)$            ▷ $\mathbf{S}_{Eh} \in \mathbb{R}^{B \times B}$, historical extrinsic similarity.
9: $\mathbf{S}_{Ef} \leftarrow \text{Sim}(\mathbf{E}_f, \tilde{\mathbf{E}}_f)$            ▷ $\mathbf{S}_{Ef} \in \mathbb{R}^{B \times B}$, future extrinsic similarity.

10: **# Compute intrinsic and extrinsic losses**
11: Let $\mathbf{I}_{\text{diag}} \in \mathbb{R}^{B \times B}$ be the identity matrix, serving as the ground-truth labels.
12: $\mathcal{L}_I \leftarrow \text{CrossEntropy}(\mathbf{S}_I, \mathbf{I}_{\text{diag}})$            ▷ Intrinsic consistency loss.
13: $\mathcal{L}_{Eh} \leftarrow \text{CrossEntropy}(\mathbf{S}_{Eh}, \mathbf{I}_{\text{diag}})$
14: $\mathcal{L}_{Ef} \leftarrow \text{CrossEntropy}(\mathbf{S}_{Ef}, \mathbf{I}_{\text{diag}})$
15: $\mathcal{L}_E \leftarrow (\mathcal{L}_{Eh} + \mathcal{L}_{Ef})/2$            ▷ extrinsic consistency loss.

16: **# Compute total training loss**
17: $\mathcal{L}_{\text{total}} \leftarrow \mathcal{L}_I + \mathcal{L}_E$            ▷ Total objective.
18: **Return:** $\mathcal{L}_{\text{total}}$

---

The core of the estimator is a stack of $N$ identical conditional Transformer blocks, architecturally similar to DiT (Peebles & Xie, 2023) but adapted to time-series tokens. Each block follows a pre-normalization design, consisting of a multi-head self-attention sub-layer and a feed-forward sub-layer. The condition embedding $\mathbf{e}_c$ is fed into a lightweight conditioning MLP to produce a conditioning signal, which is then injected into the block to modulate the token features. This yields the

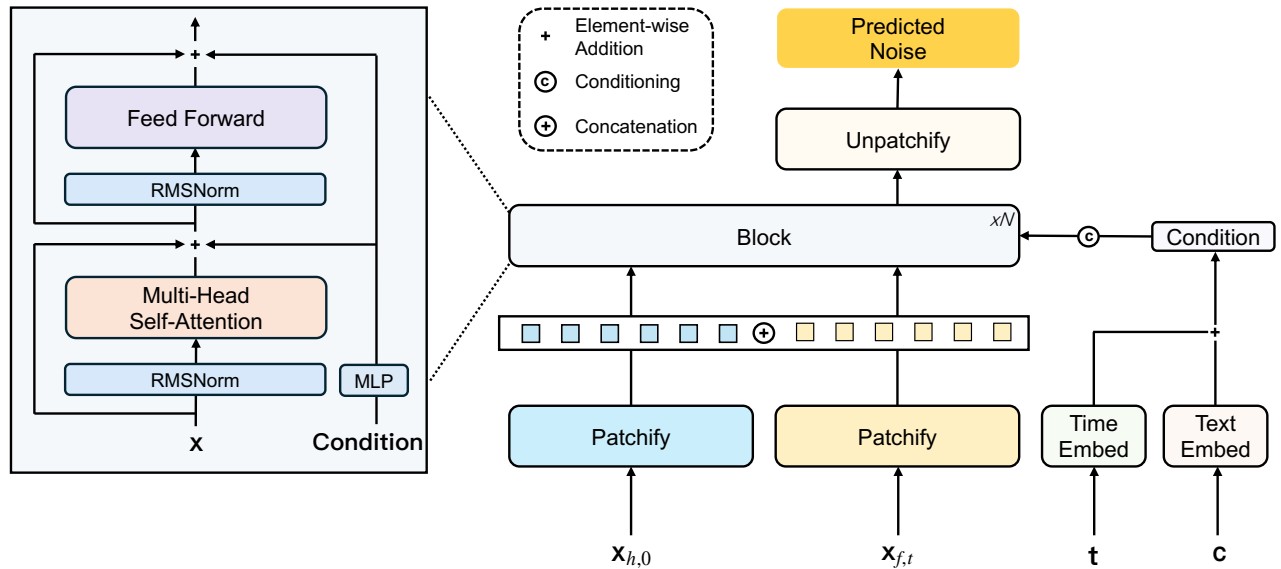

*Figure 8.* The model architecture of the noise estimator in TADIFF.

*Table 6.* The configuration of TADIFF on different datasets. $L_h$ and $L_f$ represent the history length and forecasting horizon, respectively. $\lambda_F : \lambda_A$ is the loss weight ratio of forecasting and attribution during training. $\lambda_I : \lambda_E$ is the loss weight ratio of intrinsic consistency and extrinsic consistency during finetuning.

| Type | Configuration | Synth | ETTm1 | Traffic | Exchange | Weather |
|------|---------------|-------|-------|---------|----------|---------|
| Data | $(L_h, L_f)$ | $(128, 128)$ | $(48, 48)$ | $(48, 48)$ | $(48, 48)$ | $(36, 36)$ |
| Training | Epoch | 700 | 700 | 700 | 700 | 700 |
|  | Batch size | 1024 | 1024 | 1024 | 1024 | 1024 |
|  | $\lambda_F : \lambda_A$ | $2 : 1$ | $2 : 1$ | $2 : 1$ | $2 : 1$ | $2 : 1$ |
| Finetuning | Epoch | - | 200 | 200 | 200 | 100 |
|  | Batch size | - | 256 | 256 | 256 | 256 |
|  | $\lambda_I : \lambda_E$ | - | $5 : 1$ | $20 : 1$ | $10 : 3$ | $1 : 1$ |

refined representation for the next layer: $\mathbf{H}^j = \text{Block}_j(\mathbf{H}^{j-1}, \mathbf{e}_c)$, with $\mathbf{H}^0 = \mathbf{P}$.

Finally, after all blocks, the output tokens are passed through a prediction head and then an unpatchify module to map patch features back to the original time-series space, producing the estimated noise for the future series $\hat{\boldsymbol{\epsilon}}_t \in \mathbb{R}^{L_f}$.

## D. Implementation Setting

We adopt dataset-specific configurations for each dataset, as shown in Tab. 6. Since Synth already contains diverse conditions with ground truth future time series, only the training is applied. All experiments are run on a single NVIDIA-A40 GPU with three random seeds.

## E. More Experimental Results

### E.1. Model Efficiency Analysis

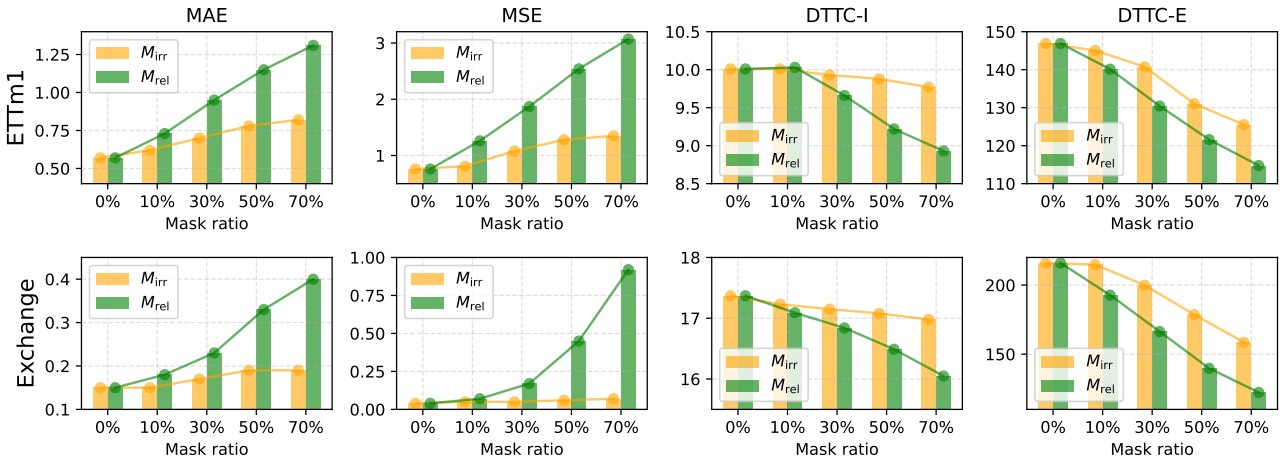

*Figure 9.* The sensitivity study of masking different ratios of relevant or irrelevant tokens in the generated text on ETTm1 and Exchange datasets. $M_{\text{irr}}$ represents masking irrelevant tokens, $M_{\text{rel}}$ represents masking relevant tokens.

In this section, we present the efficiency comparison of TADIFF with other baselines. Specifically, we compared model size and average inference time per sample on the Weather dataset using a single NVIDIA A40 GPU (batch size = 1). As shown in the Tab. 7, although our model incurs a higher inference cost than non-diffusion models, it is competitive with both foundation and diffusion base-

*Table 7.* Model size and inference time comparison on Weather dataset.

| Method | TADIFF | VerbalTS | Sundial | TimeMMD |
|---|---|---|---|---|
| Model size (MB) | 14 | 16 | 128 | 35 |
| Inference time (ms) | 340 | 463 | 272 | 17 |

lines. Given its significant forecasting improvements, the increase in model size and the reduction in inference speed are considered acceptable trade-offs.

### E.2. Sensitivity Study on Constructed Text Templates

In this section, we provide a sensitivity study to prove that TADIFF is not overfitting to the text templates in the generated text mentioned in Appendix A.3.1 but successfully captures the meaningful contents.

We manually classified the tokens appearing in the generated text into two categories: relevant and irrelevant. Relevant tokens contain meaningful semantics related to the time series, while irrelevant tokens mainly serve to maintain valid sentence structure. For example, in the sentence "The trend direction is up.", the tokens "trend", "direction", and "up" are relevant, whereas "The", "is", and "." are irrelevant. We independently masked relevant and irrelevant tokens at varying ratios and analyzed the resulting performance degradation. We conducted the experiment on the factual data of the ETTm1 and Exchange datasets, where both datasets use generated text.

As shown in Fig. 9, masking irrelevant tokens leads to significantly smaller performance drops compared with masking relevant tokens. Notably, for MAE and MSE, masking 70% of the irrelevant tokens still yields performance comparable to masking only 10% of the relevant tokens. These results demonstrate that TADiff indeed learns from the meaningful semantic information in the generated text rather than overfitting to the constructed templates.

### E.3. Conditional Decoupling Capability of Text-Attribution

In this section, we prove that the text-attribution of TADIFF can decouple the intrinsic features of the sequence from the extrinsic conditions. Specifically, we hope to verify the dependency between intrinsic features $\mathbf{x}_{h,T}$ and extrinsic conditions $\mathbf{c}_h$. We train a contrastive learning model (similar to CLIP (Radford et al., 2021)) to verify whether the relationship between $\mathbf{x}_{h,T}$ and $\mathbf{c}_h$ can be effectively captured (i.e., $\mathbf{x}_{h,T}$ is dependent of $\mathbf{c}_h$) or not ($\mathbf{x}_{h,T}$ is truly independent of $\mathbf{c}_h$).

We first use our model (TADIFF) to estimate the initial noise $\mathbf{x}_{h,T}$ based on $\mathbf{x}_{h,0}$ and $\mathbf{c}_h$. Then, we train CLIP models to learn a shared latent space between $\mathbf{x}_{h,T}$ and $\mathbf{c}_h$ through contrastive learning. We evaluate the CLIP model by measuring its accuracy in retrieving the most similar condition $\mathbf{c}_h$ from three candidates given $\mathbf{x}_{h,T}$. For comparison, we also train

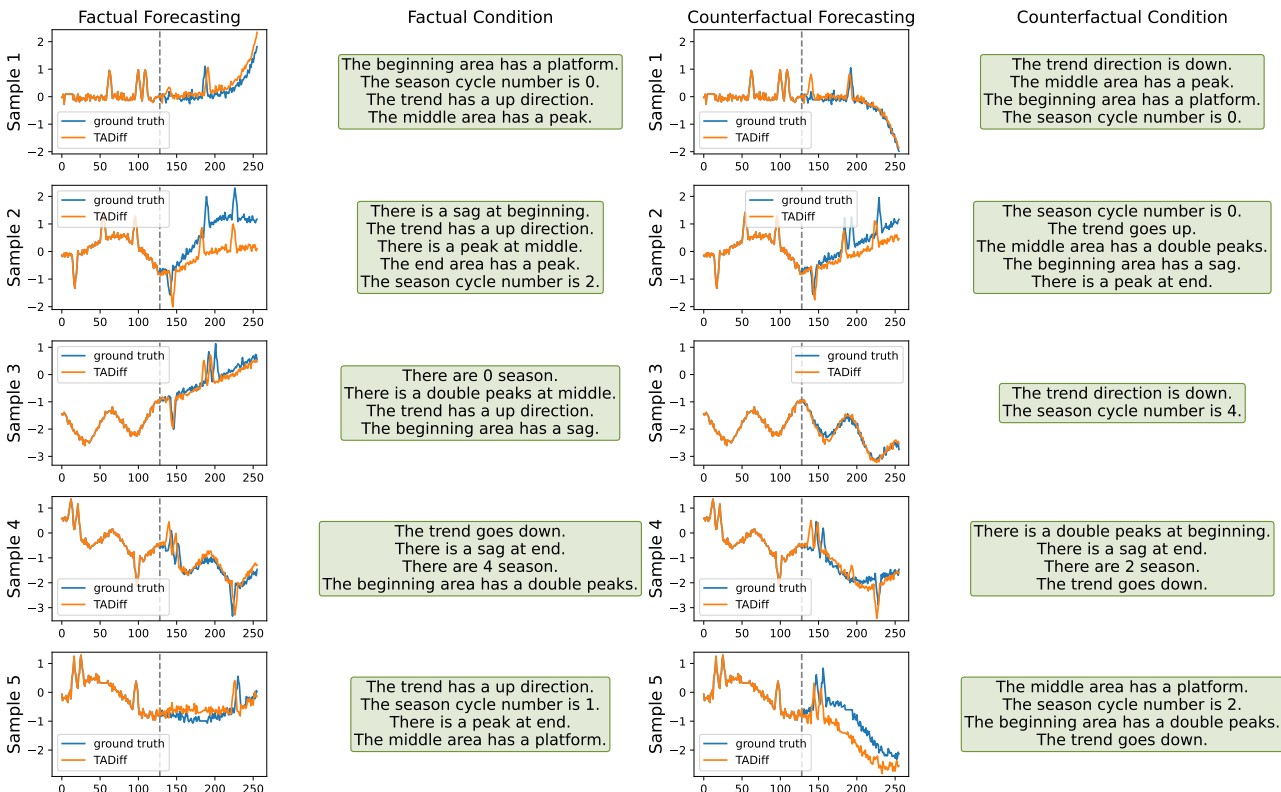

*Figure 10.* The case study of time series forecasting on Synth dataset.

additional CLIP models between $\mathbf{x}_{h,0}$ and $\mathbf{c}_h$ using the same model architecture.

The results in the following Tab. 8 prove the independence between $\mathbf{x}_{h,T}$ and $\mathbf{c}_h$. It can be observed that retrieving $\mathbf{c}_h$ given $\mathbf{x}_{h,T}$ yields significantly lower accuracy compared to retrieving $\mathbf{c}_h$ given $\mathbf{x}_{h,0}$, with the performance of the former approaching random guessing. It illustrates that the derived initial noise from the text-attribution stage has truly become independent of $\mathbf{c}_h$, decoupling the intrinsic features of the sequence from the extrinsic conditions.

*Table 8.* Accuracy of retrieving the most similar $\mathbf{c}_h$ from 3 candidates, the ideal accuracy of random guessing is approximately 33%.

| Setting | Synth | ETTm1 | Exchange | Traffic | Weather |
|---|---|---|---|---|---|
| $\mathbf{x}_{h,0} \to \mathbf{c}_h$ | 97.42 | 93.34 | 98.17 | 96.75 | 77.71 |
| $\mathbf{x}_{h,T} \to \mathbf{c}_h$ | 32.27 | 37.84 | 42.93 | 39.70 | 37.18 |

### E.4. Case Study

In this section, we provide several visualization results of TADIFF on Synth dataset, including both the factual and counterfactual forecasting. The results are presented in Fig. 10, which demonstrates that TADIFF considers both the constraint of the historical intrinsic feature and control of the future condition. Furthermore, TADIFF demonstrates strong generalization capabilities to diverse future conditions, providing accurate forecasts in both factual and counterfactual settings.

