# OpenReview forum: "What if Tomorrow is the World Cup Final? Counterfactual Time Series Forecasting with Textual Conditions"
_ICML.cc/2026/Conference — ICML 2026 regular_

### Official Review · Reviewer_F1XM · 2026-02-14

**Soundness:** 3
**Presentation:** 3
**Significance:** 3
**Originality:** 3
**Overall Recommendation:** 5
**Confidence:** 4

**Summary:**

This paper introduces a novel framework for counterfactual time series forecasting under textual conditions, aiming to model how future sequences evolve given natural language descriptions. The authors propose TADIFF, a diffusion model that disentangles intrinsic historical features from extrinsic textual conditions through a two-stage attribution and forecasting mechanism. Additionally, the paper presents a semantic evaluation framework (DTTC) to assess consistency between forecasts, historical dynamics, and textual controls. Extensive experiments on both synthetic and real-world datasets demonstrate improved numerical accuracy and semantic alignment.

**Compliance With Llm Reviewing Policy:**

Affirmed.

**Key Questions For Authors:**

1.Could the authors provide comparable visualization results on real-world datasets to better demonstrate the model’s behavior and effectiveness in practical scenarios?

2.In constructing the counterfactual dataset, the future condition is selected as the one with the highest similarity to the historical condition. Could the authors clarify whether this design may reduce the discrepancy between factual and counterfactual scenarios?

3.For unimodal datasets, textual descriptions are generated from intrinsic statistical properties (e.g., linear trend, FFT frequency) rather than genuine external events or interventions. To what extent do these experiments truly validate the model’s ability to handle extrinsic counterfactual conditions, as motivated in the introduction?

**Limitations:**

yes

**Strengths And Weaknesses:**

Strengths:

1.The paper introduces counterfactual time series forecasting with free-form textual conditions, which is meaningful and potentially impactful for real-world forecasting scenarios.

2.The experiments cover both synthetic and real-world datasets, and the proposed semantic consistency metric provides a way to evaluate forecasts even without ground truth.

Weaknesses:

1.Most qualitative visualizations are conducted on the synthetic dataset, while visual results on real-world datasets are largely missing. Including such examples would strengthen empirical credibility.

2.In constructing the counterfactual dataset, the authors select the future condition that achieves the highest similarity score with the historical condition. This may reduce the discrepancy between factual and counterfactual scenarios, potentially limiting the model’s ability to handle more extreme or substantially different real-world counterfactual conditions.

3.For unimodal datasets, the textual descriptions are constructed from intrinsic statistical properties of the time series (e.g., linear trend, FFT frequency) rather than genuine external future events or interventions as described in the motivation. Therefore, experiments on these datasets may not fully validate the model’s claimed ability to forecast under truly extrinsic counterfactual conditions.

4.The paper does not provide code at this stage, which raises concerns regarding reproducibility.

---

> ### Author Rebuttal · Authors · 2026-03-31
>
> Due to space limitations, we provide the **experimental results via the link**: https://anonymous.4open.science/api/repo/TADiff/file/res.pdf?v=531e6306. We sincerely apologize for any inconvenience caused.
>
> > **W1/Q1:** Most qualitative visualizations are conducted on the synthetic dataset, while visual results on real-world datasets are largely missing. Including such examples would strengthen empirical credibility.
>
> We agree that providing comprehensive data coverage in the qualitative analysis is important. In the visualization of the initial noise distribution, we **included both synthetic and real-world datasets**, as discussed in Finding 5 in Sec. 4.3 of our paper. The conclusion that our proposed novel attribution method improves the initial noise space is consistent across synthetic and real-world datasets.
>
> In the case study, we only visualize the results on the synthetic dataset because it is fully controlled and therefore easier for human interpretation. Nevertheless, we also add cases of forecasting results for real-world datasets in Fig. 1 of the link above. The results show that text attribution improves forecasting quality, which **is consistent with the conclusions of our paper**.
>
> > **W2/Q2:** In constructing the counterfactual dataset, the authors select the future condition that achieves the highest similarity score with the historical condition. This may reduce the discrepancy between factual and counterfactual scenarios, potentially limiting the model's ability to handle more extreme or substantially different real-world counterfactual conditions.
>
> Thanks for your valuable comments. We want to claim that the constructing of counterfactual data **have considered the difference between history and future.** Since before selecting the future condition with highest similarity, we first sample a random condition from a large range of conditions, which include many diverse conditions.
>
> We believe the counterfactual data constructed through this strategy **is more close to the real-world data distribution**, where the extreme counterfactual conditions exist but are scarce. We further count DTTC similarity between the history and future across three kinds of data: i) factual data, ii) counterfactual data through our construction, iii) counterfactual data through totally random sampling. The similarity distributions of these three kinds of data are visualized in the Fig. 2 in the link above. It can be observed that the distribution of ii) is between i) and iii), achieving a balance between diversity and realism, and also cover the extreme conditions.
>
> > **W3/Q3:** For unimodal datasets, the textual descriptions are constructed from intrinsic statistical properties of the time series (e.g., linear trend, FFT frequency) rather than genuine external future events or interventions as described in the motivation. Therefore, experiments on these datasets may not fully validate the model's claimed ability to forecast under truly extrinsic counterfactual conditions.
>
> Thanks for your valuable comments. We believe statistical features can be regarded as a **proxy for the influence of external events** on time series, meaning that statistical text provides a more direct way to describe the impact of such events. Similar methods of data construction are also adopted in time series generation [1] and forecasting [2]. As time series are known to exhibit non-stationarity, the statistical features may differ between the historical and future periods. If a model is unaware of such differences, it may fail to produce reasonable forecasts.
>
> The extracted statistical features can also be observed in real-world data with similar textual expressions. For example, in the real-world dataset Weather utilized in our experiments, experts describe temperature changes based on their trends. Therefore, we believe that the **extracted statistical features can reflect real-world external control conditions**. At the same time, this is an efficient approach to **expanding the range of available datasets**, allowing us to support more unimodal datasets without accompanied texts.
>
> [1] VerbalTS: Generating Time Series from Texts. ICML 2025.
>
> [2] DualSG: A Dual-Stream Explicit Semantic-Guided Multivariate Time Series Forecasting Framework. ACM MM 2025.
>
> > **W4:** The paper does not provide code at this stage, which raises concerns regarding reproducibility.
>
> Thanks for your suggestion. We provide a temporary anonymous link for the code: https://anonymous.4open.science/r/TADiff. We understand the reviewer's concern about reproducibility, and we promise to open-source the code, model weights, and datasets upon the acceptance of our paper, as mentioned in Sec.4 of our paper.

---

> > ### Author Rebuttal · Reviewer_F1XM · 2026-04-01
> >
> > My concerns have been adequately addressed.

---

> > > ### Author Response · Authors · 2026-04-02
> > >
> > > Thank you very much for recognizing the novelty and contributions of our work, and for raising the score of our paper. We are delighted that our responses have adequately addressed your concerns. We sincerely appreciate your valuable time, thoughtful feedback, and effort throughout the review process. Thank you once again for your support and consideration.

---

### Official Review · Reviewer_xQmL · 2026-03-08

**Soundness:** 3
**Presentation:** 4
**Significance:** 3
**Originality:** 3
**Overall Recommendation:** 4
**Confidence:** 4

**Summary:**

The authors work on a very interesting counterfactual time series forecasting question with free form textual conditions. They propose a diffusion-based model TADIFF with a text attribution mechanism and presents a novel evaluation metric DTTC for settings where ground-truth counterfactual futures are unavailable. The paper then validates their method and compares with other baseline models on both the synthetic data where they know the ground truth and the real world datasets which they rely on the proposed evaluation metric.

**Compliance With Llm Reviewing Policy:**

Affirmed.

**Key Questions For Authors:**

See my comments above:
1. Could the authors comment on my concern expressed above about the potential risk of circular evaluation?

2. How stable is DTTC across slightly different encoder architectures?

3. Does the performance of the model stay relatively robust if organic / daily natural languages are used as the textual information instead of the rigid text templates?

**Limitations:**

yes

**Strengths And Weaknesses:**

Soundness: The paper's formulation is technically coherent and well motivated. the biggest concern I currently have with the paper is that, if I am understanding the paper correctly, the DTTC model is trained on the same datasets as TADIFF and if the model is trained to optimize DTTC as a learnt metric there may be some level of gameable circular evaluation.

Presentation: The overall presentation is well-paced and clear. The visualizations are helpful.

Significance: This counterfactual forecasting question under textual conditions is timely and of huge real world impact as it advances the field from standard categorical treatment variables to unstructured textual conditions to allow the model capturing far more rich and subtle real world contexts. However, I think it seems to be a bit overselling as based on the experiments, the textual conditions are generated from the statistical features extracted from data so the model is pretty much learning the rigid text template instead of organic daily natural language.

Originality: Extending counterfactual intervention modeling from discrete / categorical treatments to free form textural information is an important and meaningful step forward.

---

> ### Author Rebuttal · Authors · 2026-03-31
>
> Due to space limitations, we provide the **experimental results via the link**: https://anonymous.4open.science/api/repo/TADiff/file/res.pdf?v=531e6306. We sincerely apologize for any inconvenience caused.
>
> > **W1:** The DTTC model is trained on the same datasets as TADIFF and if the model is trained to optimize DTTC as a learnt metric there may be some level of gameable circular evaluation.
>
> We understand the reviewer's concern and want to clarify that this is not a gameable circular evaluation for the following reasons:
> - **The training data of DTTC and TADiff are not exactly the same.** The DTTC model is trained on factual data, while TADiff is fine-tuned on constructed counterfactual data. These two types of data share a similar domain but contain different sample instances.
> - **We have proven that DTTC model is reliable.** As shown in Finding 7 in Sec. 4.3 of our paper, DTTC models effectively capture the semantic consistency. We believe optimizing a model through a reliable metric is reasonable, as the metric can truly reflect forecasting quality.
> - **We also consider numerical accuracy metrics in our evaluation to avoid gameable circular evaluation.** As discussed in Finding 3 in Sec. 4.3 of our paper, fine-tuning with DTTC scores does not degrade MAE/MSE performance, demonstrating that DTTC scores are not in conflict with numerical accuracy metrics.
>
> > **W2:** I think it seems to be a bit overselling as based on the experiments, the textual conditions are generated from the statistical features extracted from data so the model is pretty much learning the rigid text template instead of organic daily natural language.
>
> We agree that it is important to make the evaluation close to real-world textual conditions, and we believe such evaluations are already included in our experiments.
> - We consider the **real-world Weather dataset** with expert annotations in our experiments. The expert annotations in Weather are organic daily natural language, proving that our method performs well under complex real-world scenarios.
> - We proved that our method is **not learning rigid templates**, but meaningful semantics. As discussed in Appendix D.3 of our paper, we randomly masked relevant and irrelevant tokens in texts to study the influence. The results show that our method indeed learns from meaningful semantic information in the generated text rather than overfitting to the constructed templates.
>
> > **Q1:** Could the authors comment on my concern expressed above about the potential risk of circular evaluation?
>
> Thanks for your question.
>
> As discussed in the response to W1, we have demonstrated the reliability and rationality of the DTTC scores, and we believe there is no risk of circular evaluation.
>
> > **Q2:** How stable is DTTC across slightly different encoder architectures?
>
> We agree that the robustness of evaluation model is important. Therefore, we compare the influence of encoder architectures and loss functions of DTTC.
> - **Change text encoder.** We replace the original text encoder with a pretrained Long-CLIP text encoder [1] accompanied by a trainable layer.
> - **Change loss function.** We replace the contrastive loss with cross-entropy loss.
>
> As shown in Table 2 of link above, retrieval performance remains stable across different encoder architectures and loss functions, proving that the **DTTC model is robust**. We further evaluate all methods with different DTTC models in Table 3 of link above, and the conclusions remain consistent with those in our paper, where our method (TADiff) achieves the best forecasting performance.
>
> [1] Long-CLIP: Unlocking the Long-Text Capability of CLIP. ECCV 2024.
>
> > **Q3:** Does the performance of the model stay relatively robust if organic / daily natural languages are used as the textual information instead of the rigid text templates?
>
> As discussed in the response to W2, we believe we are capable of handling daily natural languages. Experiments on the real-world dataset Weather demonstrate that our method performs well in real-world scenarios. The sensitivity study by assessing the influence of relevant/irrelevant tokens in the input conditions in Appendix D.3 further shows that our method learns meaningful semantics rather than rigid templates.
>
> We conduct futher experiments to prove that **our method stays robust to daily natural language**:
> - We used GPT-4.1 to refine the templated-style texts in the Traffic dataset. We provide an example in Fig. 3 of link above to illustrate the differences before and after refinement.
> - We reconstruct our method (TADiff) and the evaluation pipeline (DTTC model) on this LLM-refined dataset. We compared TADiff with several strong baselines (including the unimodal model Sundial and the multimodal model TimeMMD).
>
> As shown in Table 5 of link above, the results indicate that TADiff still outperforms baselines on both factual and counterfactual forecasting, further demonstrating the robustness of our method on daily natural languages.

---

> > ### Author Rebuttal · Reviewer_xQmL · 2026-04-03
> >
> > Thanks for the detailed response.

---

> > > ### Author Response · Authors · 2026-04-04
> > >
> > > Thank you for acknowledging the motivation, contribution, novelty, and presentation of our paper. We are glad that we were able to adequately address all of your concerns. We sincerely appreciate your valuable time and effort in reviewing our paper, and we thank you again for your constructive feedback.

---

### Official Review · Reviewer_vSf2 · 2026-03-13

**Soundness:** 3
**Presentation:** 2
**Significance:** 3
**Originality:** 4
**Overall Recommendation:** 4
**Confidence:** 4

**Summary:**

this paper claims to contribute a counterfactual evaluation framework for text-conditioned time series forecasting using diffusion based methods. the authors also suggest a novel text-attribution mechanism that distinguishes mutable from immutable factors

**Compliance With Llm Reviewing Policy:**

Affirmed.

**Final Justification:**

The rebuttal is helpful. I reinforce my prior assessment.

**Key Questions For Authors:**

please see above

**Limitations:**

Yes

**Strengths And Weaknesses:**

while I like the counter-factual idea, which I think is essential for forecasting and has values in various applications. this paper in general is a bit hard to follow, therefore as far as I read I have a few concerns:

1. If this model is meant for forecasting, it makes an unrealistic assumption: that we already know what the future "textual conditions" will be. In real life, we don't have those descriptions ahead of time, so the model might not be very useful in a practical setting.

2. The paper describes the work as "forecasting," but it’s actually more like a "what-if" generator. This creates two big problems: 1. since these "what-if" scenarios never actually happened, there is no real-world data to compare the results against. This makes it hard to prove the model or its scoring methods actually work. 2. the authors compare their model to standard forecasting tools. Instead, they should compare it to other AI models that generate data, as those are more relevant to this specific task.

I think what-if generator on its own is not a bad idea for forecasting. unlike fact retrieval tasks where "untrue" facts are automatically false and thus counter factual, in forecasting hypothetical scenarios are meaningful but the way they are described in this paper seems confusing to me.

therefore in my opinion the title "Counterfactual Forecasting" doesn't accurately describe the whole paper. Because the research covers both regular forecasting and "what-if" generation, the current title might confuse or mislead readers about what the study is actually about. this also makes the contributions of this paper a bit confusing.

that said given these narrative issues im giving this paper a borderline rating. I rounded my rating up to weak accept provided that the authors clarify the contributions in the writing.

---

> ### Author Rebuttal · Authors · 2026-03-31
>
> > **W1:** If this model is meant for forecasting, it makes an unrealistic assumption: that we already know what the future "textual conditions" will be. In real life, we don't have those descriptions ahead of time, so the model might not be very useful in a practical setting.
>
> Thanks for the valuable comments.
>
> We believe the task formulation does not conflict with realistic assumptions. As discussed in Sec. 3.1 and 3.5 of our paper, we assume that future textual conditions are uncertain. In other words, we do not truly know what will happen in the future, but instead aim to make reasonable forecasts under a variety of potential future conditions.
>
> In practical applications, we provide the model with multiple possible conditions and generate forecasts for all of them. As illustrated in Fig. 1 of our paper, which depicts a real-world traffic flow scenario, different counterfactual forecasts can help relevant departments prepare contingency plans for emergencies in advance.
>
> > **W2:** The paper describes the work as "forecasting," but it's actually more like a "what-if" generator. This creates two big problems: 1. since these "what-if" scenarios never actually happened, there is no real-world data to compare the results against. This makes it hard to prove the model or its scoring methods actually work. 2. the authors compare their model to standard forecasting tools. Instead, they should compare it to other AI models that generate data, as those are more relevant to this specific task.
>
> This is an interesting perspective. We believe that time series generation and forecasting tasks **share many similarities in their underlying paradigms**, as both can be formulated as generating corresponding time series given certain conditions. However, forecasting places greater emphasis on the temporal horizon and relies on both historical sequences and textual conditions, whereas generation often only considers textual conditions.
>
> For the two questions raised by the reviewer:
> - **We have conducted experiments to validate the reliability of our evaluation model.** Counterfactual forecasting faces the challenge of lacking ground truth. To address this limitation, we propose evaluating semantic consistency instead of relying solely on numerical accuracy, which is commonly used in traditional forecasting tasks. In Finding 7 of our paper, we demonstrate that the evaluation model is reliable because it learns the alignment between historical and future sequences, as well as the alignment between textual conditions and sequences.
> - **The baselines we considered provide comprehensive coverage.** In addition to standard forecasting models, we compare against the counterfactual forecasting model CT [1], which uses discrete labels as conditions, and the generation method VerbalTS [2], which generates time series from text. The results show that our method achieves better consistency with both historical data and future conditions.
>
>
> > **W3:** In my opinion the title "Counterfactual Forecasting" doesn't accurately describe the whole paper. Because the research covers both regular forecasting and "what-if" generation, the current title might confuse or mislead readers about what the study is actually about. this also makes the contributions of this paper a bit confusing.
>
> Thanks for your valuable comments.
>
> We believe "Counterfactual Forecasting" accurately describes the task of our paper:
> - Counterfactual forecasting is a concept that complements factual forecasting, meaning that counterfactual scenarios are defined relative to factual ones. Therefore, the task name "Counterfactual Forecasting" implicitly includes factual forecasting.
> - We follow the existing definition of counterfactual forecasting used in related works [1,3], where both factual and counterfactual evaluations are included. However, these works attempt to construct ground truth for counterfactual settings, which conflicts with the inherent uncertainty of counterfactual scenarios. We discussed the limitations of their data construction in Sec. 2.2 of our paper.
>
> We want to reiterate our contributions to avoid potential confusion:
> - We follow the existing paradigm of counterfactual tasks and are the first to address counterfactual time series forecasting with complex textual conditions.
> - We propose a comprehensive evaluation framework that combines numerical accuracy and semantic alignment measures, resolving the dilemma of ground truth absence in counterfactual settings.
> - We develop a text-attributive time series diffusion model that disentangles intrinsic historical patterns from textual context, along with a training strategy that improves the s adaptability to diverse counterfactual conditions through the constructed counterfactual data.
>
> [1] Causal Transformer for Estimating Counterfactual Outcomes. ICML 2022.
>
> [2] VerbalTS: Generating Time Series from Texts. ICML 2025.
>
> [3] Counterfactual Generative Models for Time-Varying Treatments. KDD 2024.

---

> > ### Author Rebuttal · Reviewer_vSf2 · 2026-04-04
> >
> > Thank you for your detailed rebuttal. I retain my current assessment.

---

> > > ### Author Response · Authors · 2026-04-04
> > >
> > > Thank you for your recognition of the motivation and contributions of our paper. We are glad to adequately address all of your concerns. Thank you again for your valuable time, effort, and thoughtful feedback throughout the review process.

---

### Official Review · Reviewer_4eV7 · 2026-03-15

**Soundness:** 2
**Presentation:** 2
**Significance:** 2
**Originality:** 3
**Overall Recommendation:** 3
**Confidence:** 4

**Summary:**

This paper studies counterfactual time series forecasting conditioned on textual descriptions of future events, where forecasts must account for both historical dynamics and hypothetical future conditions. The authors aim to explore a fundamental problem: how time series may evolve under complex and stochastic future scenarios that are not observed in the training data. To address this, the paper proposes TADIFF, a text-attributive diffusion model that disentangles intrinsic historical patterns from extrinsic textual conditions through a text-attribution mechanism and then generates forecasts conditioned on these factors.

Overall, the submission's key domain is multimodal time series forecasting and counterfactual modeling. In addition to the modeling approach, the paper introduces a semantic evaluation framework (DTTC) that measures the consistency between generated forecasts, historical patterns, and textual conditions, allowing evaluation even when counterfactual ground-truth sequences are unavailable. Experiments on synthetic and real-world datasets show that the proposed method achieves improved numerical forecasting accuracy and stronger semantic alignment compared with several unimodal and multimodal baselines.

**Compliance With Llm Reviewing Policy:**

Affirmed.

**Key Questions For Authors:**

In several datasets, textual descriptions are generated from extracted statistical features or predefined templates. How representative are these texts of realistic future events, and how would the method perform with naturally occurring textual conditions (e.g., real news or event descriptions)?

The DTTC metric relies on a learned encoder to measure semantic consistency between forecasts and textual conditions. How sensitive are the evaluation results to the design or training of the DTTC model, and could different encoders lead to different conclusions?

The proposed attribution mechanism is central to the method. Could the authors provide deeper analysis or visualization demonstrating how intrinsic features and textual conditions are disentangled in practice?

How well does the model generalize when future textual conditions differ significantly from those seen during training (e.g., entirely new types of events or narratives)?

**Limitations:**

yes

**Strengths And Weaknesses:**

Pros

The paper introduces the task of counterfactual time series forecasting with textual conditions, which extends traditional forecasting to “what-if” scenarios and allows modeling of complex future events described in natural language.

The proposed TADIFF model combines diffusion-based generative forecasting with a text-attribution mechanism that disentangles intrinsic historical dynamics from extrinsic textual conditions.

The paper proposes the DTTC metric to assess semantic consistency between forecasts, historical patterns, and textual conditions, addressing the challenge of evaluating counterfactual forecasts without ground truth.

Experiments on both synthetic and real-world datasets show improvements in forecasting accuracy and semantic alignment compared to several unimodal and multimodal baselines.

Cons

For several datasets, textual descriptions are automatically generated from extracted features or templates, which may limit the realism and diversity of the multimodal conditions.

While DTTC provides a semantic consistency measure, its reliability depends on the learned encoder and contrastive training setup, which may introduce evaluation bias.

The proposed diffusion-based framework with additional attribution and fine-tuning stages increases model complexity and computational cost compared to simpler forecasting approaches.

The practical relevance of counterfactual textual conditions in real forecasting applications is not fully demonstrated beyond synthetic or constructed scenarios.

---

> ### Author Rebuttal · Authors · 2026-03-31
>
> Due to space limitations, we provide the **experimental results via the link**: https://anonymous.4open.science/api/repo/TADiff/file/res.pdf?v=531e6306. We apologize for any inconvenience caused.
>
> > **W1:** Texts from extracted features may limit the realism and diversity.
> >
> > **Q1:** How would the method perform with natural texts.
>
> We understand the concern regarding the constructed text, and we adopted several strategies to ensure the realism and diversity:
> - **The extracted features are realistic.** Similar features are also described in real-world datasets. For instance, *temperature change* in Weather dataset shares similar semantics with the *trend* feature. Meanwhile, the template does not compromise realism, as similar structural expressions also exist in real texts.
> - **The constructed texts are diverse:**
>   - We design various text templates.
>   - We randomly shuffle the description order of features.
>
> We believe this is an effective way to expand the range of available datasets. Meanwhile, dataset (Weather) with natural texts is included in experiments, and the results demonstrate our method's capability to **handle natural texts.**
>
> > **W2:** DTTC's reliability depends on learned encoder and contrastive training.
> >
> > **Q2:** How sensitive are evaluations to the design or training of DTTC?
>
> Thanks for your valuable comments.
>
> We believe the DTTC is reliable and fair for the following reasons:
> - **Model-based evaluation is a mature approach,** many works [1,2] have adopted similar evaluation method. Inspired by this, we apply it to the counterfactual forecasting, since this task focus more on semantic consistency, as discussed in Sec. 3.4 of paper.
> - **We proved the DTTC model is reliable.** As discussed in Finding 7 and Table 4 in our paper, DTTC models achieve high retrieval accuracy, demonstrating their ability to align semantics.
>
> We understand the concern regarding sensitivity and modify DTTC model with different designs and training:
> - **Change text encoder.** We replace the original text encoder with pretrained Long-CLIP text encoder accompanied by a trainable layer.
> - **Change loss function.** We replace the contrastive loss with cross-entropy loss.
>
> As shown in Table 2 of link, retrieval performance remains stable across different designs and training, proving that **DTTC model is robust**. We further evaluate all methods with different DTTC models in Table 3 of link, and **conclusions remain consistent** with those in our paper, where our method achieves the best forecasting.
>
> [1] CLIPScore: A Reference-free Evaluation Metric for Image Captioning. EMNLP 2021.
>
> [2] VerbalTS: Generating Time Series from Texts. ICML 2025.
>
> > **W3:** The proposed attribution and fine-tuning increase model complexity and computational cost.
>
> We understand the concern about complexity and cost.
>
> For **model complexity**, we want to clarify it **has not increased** compared to simpler forecasting methods. Attribution and forecasting are based on the exact same model, no additional parameters are introduced.
>
> For **computational cost**, we believe the increase **is acceptable** compared to performance improvement:
> - Fine-tuning requires far fewer training epochs (200) than pre-training (700).
> - Even with attribution, our method's inference speed remains competitive with baselines, as shown in Table 1 of the link.
>
> > **W4:** The relevance of counterfactual conditions in real applications is not fully demonstrated beyond synthetic or constructed scenarios.
>
> Thanks for the valuable comments. We would like to clarify that we **follow the common strategy in related work [3]** to construct counterfactual conditions. These synthetic or constructed scenarios represent conditions that do not actually occur in real world but could plausibly happen. Our experiments cover **a wide range of real-world datasets**, including energy, traffic and weather. Results demonstrate our method is capable of handling complex real-world cases.
>
> [3] Causal Transformer for Estimating Counterfactual Outcomes. ICML 2022.
>
> > **Q3:** Deeper analysis about how intrinsic features and textual conditions are disentangled?
>
> In Appendix D.4 of paper, **we demonstrate that text-attribution can disentangle extrinsic conditions from intrinsic features.** We compare the classification accuracy on time series before and after attribution. Results show that sequence after attribution has low relevance to texutal conditions.
>
> > **Q4:** Could model generalize to future events unseen in training?
>
> We agree that evaluating generalization to unseen events is valuable.
>
> We count the number of event types across different splits of Weather dataset and find 674 samples (~16%) involve new event types that only appear in test split. Event types are defined as combinations of attributes. We evaluate all methods on these samples and report the results in Table 4 of the link. Results show that our model outperforms baselines and **is capable of handling new events.**

---

### Decision · Program_Chairs · 2026-04-30

**Decision:**

Accept (regular)

**Comment:**

This paper introduces counterfactual time series forecasting with textual conditions, proposing TADIFF, a diffusion model with a text-attribution mechanism that disentangles intrinsic historical patterns from extrinsic textual conditions, along with DTTC, a semantic evaluation metric for settings without counterfactual ground truth.

The paper received 4 reviews and the reviewers unanimously viewed the work as timely and original, and appreciated extending counterfactual forecasting to unstructured textual conditions, as well as the empirical gains reported on both synthetic and real-world datasets. Despite this, there were several major points of concern that the reviewers pointed out:

- The realism and diversity of the automatically generated textual conditions.
- The reliability and potential risk of circular evaluation of the DTTC metric.
- The framing of the problem as forecasting versus “what-if” generation.

The rebuttal provided was detailed, and the authors did a good job in addressing the above points through DTTC robustness analyses, further experiments on LLM-refined datasets, and additional visualizations, with 3 reviewers explicitly confirming that their concerns were resolved. Given the generally positive reviews and adequate rebuttal, I recommend accepting the paper, and request the authors to integrate the new results and visualizations into the camera-ready version of the paper, clarify the task framing, and release code and data as promised.